# Heuristic-Guided Reinforcement Learning

**Ching-An Cheng**
Microsoft Research
Redmond, WA
chinganc@microsoft.com

**Andrey Kolobov**
Microsoft Research
Redmond, WA
akolobov@microsoft.com

**Adith Swaminathan**
Microsoft Research
Redmond, WA
adswamin@microsoft.com

## Abstract

We provide a framework for accelerating reinforcement learning (RL) algorithms by heuristics constructed from domain knowledge or offline data. Tabula rasa RL algorithms require environment interactions or computation that scales with the horizon of the sequential decision-making task. Using our framework, we show how heuristic-guided RL induces a much shorter-horizon subproblem that provably solves the original task. Our framework can be viewed as a horizon-based regularization for controlling bias and variance in RL under a finite interaction budget. On the theoretical side, we characterize properties of a good heuristic and its impact on RL acceleration. In particular, we introduce the novel concept of an improvable heuristic, a heuristic that allows an RL agent to extrapolate beyond its prior knowledge. On the empirical side, we instantiate our framework to accelerate several state-of-the-art algorithms in simulated robotic control tasks and procedurally generated games. Our framework complements the rich literature on warm-starting RL with expert demonstrations or exploratory datasets, and introduces a principled method for injecting prior knowledge into RL.

## 1 Introduction

Many recent empirical successes of reinforcement learning (RL) require solving problems with very long decision-making horizons. OpenAI Five [1] used episodes that were 20000 timesteps on average, while AlphaStar [2] used roughly 5000 timesteps. Long-term credit assignment is a very challenging statistical problem, with the sample complexity growing quadratically (or worse) with the horizon [3]. Long horizons (or, equivalently, large discount factors) also increase RL's computational burden, leading to slow optimization convergence [4]. This makes RL algorithms require prohibitively large amounts of interactions and compute: even with tuned hyperparameters, AlphaStar needed over $10^8$ samples and OpenAI Five needed over $10^7$ PFLOPS of compute.

A popular approach to mitigate the statistical and computational issues of tabula rasa RL methods is to warm-start or regularize learning with prior knowledge [1, 2, 5–10]. For instance, AlphaStar learned a policy and value function from human demonstrations and regularized the RL agent using imitation learning (IL). AWAC [9] warm-started a policy using batch policy optimization on exploratory datasets. While these approaches have been effective in different domains, none of them explicitly address RL's complexity dependence on horizon.

In this paper, we propose a complementary regularization technique that relies on heuristic value functions, or *heuristics*[1] for short, to effectively shorten the problem horizon faced by an online RL agent for fast learning. We call this approach Heuristic-Guided Reinforcement Learning (HuRL). The core idea is simple: given a Markov decision process (MDP) $\mathcal{M} = (\mathcal{S}, \mathcal{A}, P, r, \gamma)$ and a heuristic $h : \mathcal{S} \to \mathbb{R}$, we select a mixing coefficient $\lambda \in [0, 1]$ and have the agent solve a new MDP $\widetilde{\mathcal{M}} = (\mathcal{S}, \mathcal{A}, P, \widetilde{r}, \widetilde{\gamma})$ with a reshaped reward and a smaller discount (i.e. a shorter horizon):

$$\widetilde{r}(s,a) := r(s,a) + (1-\lambda)\gamma\mathbb{E}_{s'\sim P(\cdot|s,a)}[h(s')] \quad \text{and} \quad \widetilde{\gamma} := \lambda\gamma. \qquad (1)$$

---

[1]We borrow this terminology from the planning literature to refer to guesses of $V^*$ in an MDP [11].

35th Conference on Neural Information Processing Systems (NeurIPS 2021).

HuRL effectively introduces horizon-based regularization that determines whether long-term value information should come from collected experiences or the heuristic. By modulating the effective horizon via $\lambda$, we trade off the bias and the complexity of solving the reshaped MDP. HuRL with $\lambda = 1$ recovers the original problem and with $\lambda = 0$ creates an easier contextual bandit problem [12].

A heuristic $h$ in HuRL represents a prior guess of the desired long-term return of states, which ideally is the optimal value function $V^*$ of the unknown MDP $\mathcal{M}$. When the heuristic $h$ captures the state ordering of $V^*$ well, conceptually, it becomes possible to make good long-term decisions by short-horizon planning or even acting greedily. How do we construct a good heuristic? In the planning literature, this is typically achieved by solving a relaxation of the original problem [13–15]. Alternatively, one can learn it from batch data collected by exploratory behavioral policies (as in offline RL [16]) or from expert policies (as in IL [17]).[2] For some dense reward problems, a zero heuristic can be effective in reducing RL complexity, as exploited by the guidance discount framework [18–23]. In this paper, we view heuristics as a unified representation of various forms of prior knowledge, such as expert demonstrations, exploratory datasets, and engineered guidance.

Although the use of heuristics to accelerate search has been popular in planning and control algorithms, e.g., A* [24], MCTS [25], and MPC [7, 26–28], its theory is less developed for settings where the MDP is *unknown*. The closest work in RL is potential-based reward shaping (PBRS) [29], which reshapes the reward into $\bar{r}(s, a) = r(s, a) + \gamma \mathbb{E}_{s'|s,a}[h(s')] - h(s)$ while keeping the original discount. PBRS can use any heuristic to reshape the reward while preserving the ordering of policies. However, giving PBRS rewards to an RL algorithm does not necessarily lead to faster learning, because the base RL algorithm would still seek to explore to resolve long-term credit assignment. HuRL allows common RL algorithms to leverage the short-horizon potential provided by a heuristic to learn faster.

In this work, we provide a theoretical foundation of HuRL to enable adopting heuristics and horizon reduction for accelerating RL, combining advances from the PBRS and the guidance discount literatures. On the theoretical side, we derive a bias-variance decomposition of HuRL's horizon-based regularization in order to characterize the solution quality as a function of $\lambda$ and $h$. Using this insight, we provide sufficient conditions for achieving an effective trade-off, including properties required of a base RL algorithm that solves the reshaped MDP $\widetilde{\mathcal{M}}_\lambda$. Furthermore, we define the novel concept of an *improvable* heuristic and prove that good heuristics for HuRL can be constructed from data using existing *pessimistic* offline RL algorithms (such as pessimistic value iteration [30, 31]).

The effectiveness of HuRL depends on the heuristic quality, so we design HuRL to employ a sequence of mixing coefficients (i.e. $\lambda$s) that increases as the agent gathers more data from the environment. Such a strategy induces a learning curriculum that enables HuRL to remain robust to non-ideal heuristics. HuRL starts off by guiding the agent's search direction with a heuristic. As the agent becomes more experienced, it gradually removes the guidance and lets the agent directly optimize the true long-term return. We empirically validate HuRL in MuJoCo [32] robotics control problems and Procgen games [33] with various heuristics and base RL algorithms. The experimental results demonstrate the versatility and effectiveness of HuRL in accelerating RL algorithms.

## 2 Preliminaries

### 2.1 Notation

We focus on discounted infinite-horizon Markov Decision Processes (MDPs) for ease of exposition. The technique proposed here can be extended to other MDP settings.[3] A discounted infinite-horizon MDP is denoted as a 5-tuple $\mathcal{M} = (\mathcal{S}, \mathcal{A}, P, r, \gamma)$, where $\mathcal{S}$ is the state space, $\mathcal{A}$ is the action space, $P(s'|s, a)$ is the transition dynamics, $r(s, a)$ is the reward function, and $\gamma \in [0, 1)$ is the discount factor. Without loss of generality, we assume $r : \mathcal{S} \times \mathcal{A} \to [0, 1]$. We allow the state and action spaces $\mathcal{S}$ and $\mathcal{A}$ to be either discrete or continuous. Let $\Delta(\cdot)$ denote the space of probability distributions. A decision-making policy $\pi$ is a conditional distribution $\pi : \mathcal{S} \to \Delta(\mathcal{A})$, which can be deterministic. We define some shorthand for writing expectations: For a state distribution $d \in \Delta(\mathcal{S})$ and a function $V : \mathcal{S} \to \mathbb{R}$, we define $V(d) := \mathbb{E}_{s \sim d}[V(s)]$; similarly, for a policy $\pi$ and a function $Q : \mathcal{S} \times \mathcal{A} \to \mathbb{R}$, we define $Q(s, \pi) := \mathbb{E}_{a \sim \pi(\cdot|s)}[Q(s, a)]$. Lastly, we define $\mathbb{E}_{s'|s,a} := \mathbb{E}_{s' \sim P(\cdot|s,a)}$.

---

[2]We consider the RL setting for imitation where we suppose the rewards of expert trajectories are available.

[3]The results here can be readily applied to finite-horizon MDPs; for other infinite-horizon MDPs, we need further, e.g., mixing assumptions for limits to exist.

Central to solving MDPs are the concepts of value functions and average distributions. For a policy $\pi$, we define its state value function $V^\pi$ as $V^\pi(s) := \mathbb{E}_{\rho_s^\pi}\left[\sum_{t=0}^\infty \gamma^t r(s_t, a_t)\right]$, where $\rho_s^\pi$ denotes the trajectory distribution of $s_0, a_0, s_1, \dots$ induced by running $\pi$ starting from $s_0 = s$. We define the state-action value function (or the Q-function) as $Q^\pi(s, a) := r(s, a) + \gamma \mathbb{E}_{s'|s,a}[V^\pi(s')]$. We denote the optimal policy as $\pi^*$ and its state value function as $V^* := V^{\pi^*}$. Under the assumption that rewards are in $[0, 1]$, we have $V^\pi(s), Q^\pi(s, a) \in [0, \frac{1}{1-\gamma}]$ for all $\pi$, $s \in \mathcal{S}$, and $a \in \mathcal{A}$. We denote the initial state distribution of interest as $d_0 \in \Delta(\mathcal{S})$ and the state distribution of policy $\pi$ at time $t$ as $d_t^\pi$, with $d_0^\pi = d_0$. Given $d_0$, we define the average state distribution of a policy $\pi$ as $d^\pi := (1 - \gamma)\sum_{t=0}^\infty \gamma^t d_t^\pi$. With a slight abuse of notation, we also write $d^\pi(s, a) := d^\pi(s)\pi(a|s)$.

## 2.2 Setup: Reinforcement Learning with Heuristics

We consider RL with prior knowledge expressed in the form of a heuristic value function. The goal is to find a policy $\pi$ that has high return through interactions with an unknown MDP $\mathcal{M}$, i.e., $\max_\pi V^\pi(d_0)$. While the agent here does not fully know $\mathcal{M}$, we suppose that, before interactions start the agent is provided with a heuristic $h : \mathcal{S} \to \mathbb{R}$ which the agent can query throughout learning.

The heuristic $h$ represents a prior guess of the optimal value function $V^*$ of $\mathcal{M}$. Common sources of heuristics are domain knowledge as typically employed in planning, and logged data collected by exploratory or by expert behavioral policies. In the latter, a heuristic guess of $V^*$ can be computed from the data by offline RL algorithms. For instance, when we have trajectories of an expert behavioral policy, Monte-Carlo regression estimate of the observed returns may be a good guess of $V^*$.

Using heuristics to solve MDP problems has been popular in planning and control, but its usage is rather limited in RL. The closest provable technique in RL is PBRS [29], where the reward is modified into $\bar{r}(s, a) := r(s, a) + \gamma \mathbb{E}_{s'|s,a}[h(s')] - h(s)$. It can be shown that this transformation does not introduce bias into the policy ordering, and therefore solving the new MDP $\overline{\mathcal{M}} := (\mathcal{S}, \mathcal{A}, P, \bar{r}, \gamma)$ would yield the same optimal policy $\pi^*$ of $\mathcal{M}$.

Conceptually when the heuristic is the optimal value function $h = V^*$, the agent should be able to find the optimal policy $\pi^*$ of $\mathcal{M}$ by acting myopically, as $V^*$ already contains all necessary long-term information for good decision making. However, running an RL algorithm with the PBRS reward (i.e. solving $\overline{\mathcal{M}} := (\mathcal{S}, \mathcal{A}, P, \bar{r}, \gamma)$) does not take advantage of this shortcut. To make learning efficient, we need to also let the base RL algorithm know that acting greedily (i.e., using a smaller discount) with the shaped reward can yield good policies. An intuitive idea is to run the RL algorithm to maximize $\overline{V}_\lambda^\pi(d_0)$, where $\overline{V}_\lambda^\pi$ denotes the value function of $\pi$ in an MDP $\overline{\mathcal{M}}_\lambda := (\mathcal{S}, \mathcal{A}, P, \bar{r}, \lambda\gamma)$ for some $\lambda \in [0, 1]$. However this does not always work. For example, when $\lambda = 0$, $\max_\pi \overline{V}_\lambda^\pi(d_0)$ only optimizes for the initial states $d_0$, but obviously the agent is going to encounter other states in $\mathcal{M}$. We next propose a provably correct version, HuRL, to leverage this short-horizon insight.

# 3 Heuristic-Guided Reinforcement Learning

We propose a general framework, HuRL, for leveraging heuristics to accelerate RL. In contrast to tabula rasa RL algorithms that attempt to directly solve the long-horizon MDP $\mathcal{M}$, HuRL uses a heuristic to guide the agent in solving a sequence of short-horizon MDPs so as to amortize the complexity of long-term credit assignment. In effect, HuRL creates a heuristic-based learning curriculum to help the agent learn faster.

## 3.1 Algorithm

HuRL takes a reduction-based approach to realize the idea of heuristic guidance. As summarized in Algorithm 1, HuRL takes a heuristic $h : \mathcal{S} \to \mathbb{R}$ and a base RL algorithm $\mathcal{L}$ as input, and outputs an approximately optimal policy for the original MDP $\mathcal{M}$. During training, HuRL iteratively runs the base algorithm $\mathcal{L}$ to collect data from the MDP $\mathcal{M}$ and then uses the heuristic $h$ to modify the agent's collected experiences. Namely, in iteration $n$, the agent interacts with the original MDP $\mathcal{M}$ and saves the raw transition tuples[4] $\mathcal{D}_n = \{(s, a, r, s')\}$ (line 2). HuRL then defines a reshaped MDP $\widetilde{\mathcal{M}}_n := (\mathcal{S}, \mathcal{A}, P, \widetilde{r}_n, \widetilde{\gamma}_n)$ (line 3) by changing the rewards and lowering the discount factor:

$$\widetilde{r}_n(s, a) := r(s, a) + (1 - \lambda_n)\gamma \mathbb{E}_{s'|s,a}[h(s')] \qquad \text{and} \qquad \widetilde{\gamma}_n := \lambda_n \gamma, \tag{2}$$

---

[4]If $\mathcal{L}$ learns only with trajectories, we transform each tuple and assemble them to get the modified trajectory.

**Algorithm 1** Heuristic-Guided Reinforcement Learning (HuRL)

---

**Require:** MDP $\mathcal{M} = (\mathcal{S}, \mathcal{A}, P, r, \gamma)$, RL algorithm $\mathcal{L}$, heuristic $h$, mixing coefficients $\{\lambda_n\}$.

1: **for** $n = 1, \ldots, N$ **do**
2:     $\mathcal{D}_n \leftarrow \mathcal{L}.\text{CollectData}(\mathcal{M})$
3:     Get $\lambda_n$ from $\{\lambda_n\}$ and construct $\widetilde{\mathcal{M}}_n = (\mathcal{S}, \mathcal{A}, P, \widetilde{r}_n, \widetilde{\gamma}_n)$ according to (2) using $h$ and $\lambda_n$
4:     $\pi_n \leftarrow \mathcal{L}.\text{Train}(\mathcal{D}_n, \widetilde{\mathcal{M}}_n)$
5: **end for**
6: **return** $\pi_N$

---

where $\lambda_n \in [0, 1]$ is the mixing coefficient. The new discount $\widetilde{\gamma}_n$ effectively gives $\widetilde{\mathcal{M}}_n$ a shorter horizon than $\mathcal{M}$'s, while the heuristic $h$ is blended into the new reward in (2) to account for the missing long-term information. We call $\widetilde{\gamma}_n = \lambda_n \gamma$ in (2) the *guidance discount* to be consistent with prior literature [20], which can be viewed in terms of our framework as using a zero heuristic. In the last step (line 4), HuRL calls the base algorithm $\mathcal{L}$ to perform updates with respect to the reshaped MDP $\widetilde{\mathcal{M}}_n$. This is realized by *1)* setting the discount factor used in $\mathcal{L}$ to $\widetilde{\gamma}_n$, and *2)* setting the sampled reward to $r + (\gamma - \widetilde{\gamma}_n)h(s')$ for every transition tuple $(s, a, r, s')$ collected from $\mathcal{M}$. We remark that the base algorithm $\mathcal{L}$ in line 2 always collects trajectories of lengths proportional to the original discount $\gamma$, while internally the optimization is done with a lower discount $\widetilde{\gamma}_n$ in line 4.

Over the course of training, HuRL repeats the above steps with a sequence of increasing mixing coefficients $\{\lambda_n\}$. From (2) we see that as the agent interacts with the environment, the effects of the heuristic in MDP reshaping decrease and the effective horizon of the reshaped MDP increases.

## 3.2 HuRL as Horizon-based Regularization

We can think of HuRL as introducing a horizon-based *regularization* for RL, where the regularization center is defined by the heuristic and its strength diminishes as the mixing coefficient increases. As the agent collects more experiences, HuRL gradually removes the effects of regularization and the agent eventually optimizes for the original MDP.

HuRL's regularization is designed to reduce learning variance, similar to the role of regularization in supervised learning. Unlike the typical weight decay imposed on function approximators (such as the agent's policy or value networks), our proposed regularization leverages the structure of MDPs to regulate the complexity of the MDP the agent faces, which scales with the MDP's discount factor (or, equivalently, the horizon). When the guidance discount $\widetilde{\gamma}_n$ is lower than the original discount $\gamma$ (i.e. $\lambda_n < 1$), the reshaped MDP $\widetilde{\mathcal{M}}_n$ given by (2) has a shorter horizon and requires fewer samples to solve. However, the reduced complexity comes at the cost of bias, because the agent is now incentivized toward maximizing the performance with respect to the heuristic rather than the original long-term returns of $\mathcal{M}$. In the extreme case of $\lambda_n = 0$, HuRL would solve a zero-horizon contextual bandit problem with contexts (i.e. states) sampled from $d^\pi$ of $\mathcal{M}$.

## 3.3 A Toy Example

We illustrate this idea in a chain MDP environment in Fig. 1. The optimal policy $\pi^*$ for this MDP's original $\gamma = 0.9$ always picks action $\rightarrow$, as shown in Fig. 1b-(1), giving the optimal value $V^*$ in Fig. 1a-(2). Suppose we used a smaller guidance discount $\widetilde{\gamma} = 0.5\gamma$ to accelerate learning. This is equivalent to HuRL with a zero heuristic $h = 0$ and $\lambda = 0.5$. Solving this reshaped MDP yields a policy $\widetilde{\pi}^*$ that acts very myopically in the original MDP, as shown in Fig. 1b-(2); the value function of $\widetilde{\pi}^*$ in the original MDP is visualized in Fig. 1a-(4).

Now, suppose we use Fig. 1a-(4) as a heuristic in HuRL instead of $h = 0$. This is a bad choice of heuristic (Bad $h$) as it introduces a large bias with respect to $V^*$ (cf. Fig. 1a-(2)). On the other hand, we can roll out a random policy in the original MDP and use its value function as the heuristic (Good $h$), shown in Fig. 1a-(3). Though the random policy has an even *lower* return at the initial state $s = 3$, it gives a *better* heuristic because this heuristic shares the same trend as $V^*$ in Fig. 1a-(1). HuRL run with Good $h$ and Bad $h$ yields policies in Fig. 1b-(3,4), and the quality of the resulting solutions in the original MDP, $V_\lambda^{\widetilde{\pi}^*}(d_0)$, is reported in Fig. 1c for different $\lambda$. Observe that HuRL with a good heuristic can achieve $V^*(d_0)$ with a much smaller horizon $\lambda \leq 0.5$. Using a bad $h$ does not lead to $\pi^*$ at all when $\lambda = 0.5$ (Fig. 1b-(4)) but is guaranteed to do so when $\lambda$ converges to 1. (Fig. 1b-(5)).

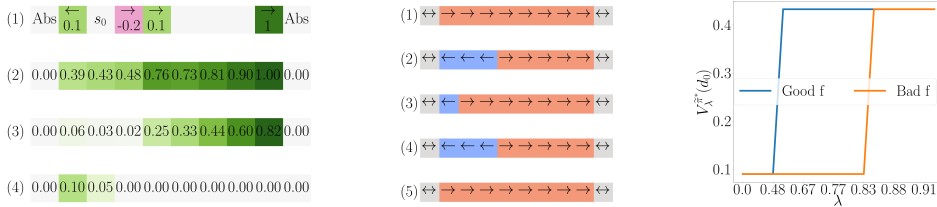

(a) Heatmap of different values.   (b) Different policy behaviors.   (c) HuRL with different $h$ and $\lambda$.

Figure 1: **Example of HuRL in a chain MDP.** Each cell in a row in each diagram represents a state from $\mathcal{S} = \{1, \ldots, 10\}$. The agent starts at state 3 ($s_0$), and states 1 and 10 are absorbing (*Abs* in subfigure a-(1)). Actions $\mathcal{A} = \{\leftarrow, \rightarrow\}$ move the agent left or right in the chain unless the agent is in an absorbing state. **Subfig. a-(1)** shows the reward function: $r(2, \leftarrow) = 0.1, r(4, \rightarrow) = -0.2, r(5, \rightarrow) = 0.1$, and all state-action pairs not shown in a-(1) yield $r = 0$. **Subfig. a-(2)** shows $V^*$ for $\gamma = 0.9$. **Subfig. a-(3)** shows a good heuristic $h$ — $V(\text{random } \pi)$. **Subfig. a-(4)** shows a bad heuristic $h$ — $V(\text{myopic } \pi)$. **Subfig. b-(1)**: $\pi^*$ for $V^*$ from a-(2). **Subfig. b-(2)**: $\tilde{\pi}^*$ from HuRL with $h = 0, \lambda = 0.5$. **Subfig. b-(3)**: $\tilde{\pi}^*$ from HuRL with the good $h$ from (a).(3) and $\lambda = 0.5$. **Subfig. b-(4)**: $\tilde{\pi}^*$ from the bad $h$ from a-(4), $\lambda = 0.5$. **Subfig. b-(5)**: $\tilde{\pi}^*$ from the bad $h$ and $\lambda = 1$. **Subfig. (c) illustrates the takeaway message**: *using HuRL with a good $h$ can find $\pi^*$ from $s_0$ even with a small $\lambda$ (see the x-axis), while HuRL with a bad $h$ requires a much higher $\lambda$ to discover $\pi^*$.*

## 4 Theoretical Analysis

When can HuRL accelerate learning? Similar to typical regularization techniques, the horizon-based regularization of HuRL leads to a bias-variance decomposition that can be optimized for better finite-sample performance compared to directly solving the original MDP. However, a non-trivial trade-off is possible only when the regularization can bias the learning toward a good direction. In HuRL's case this is determined by the heuristic, which resembles a prior we encode into learning.

In this section we provide HuRL's theoretical foundation. We first describe the bias-variance trade-off induced by HuRL. Then we show how suboptimality in solving the reshaped MDP translates into performance in the original MDP, and identify the assumptions HuRL needs the base RL algorithm to satisfy. In addition, we explain how HuRL relates to PBRS, and characterize the quality of heuristics and sufficient conditions for constructing good heuristics from batch data using offline RL.

For clarity, we will focus on the reshaped MDP $\widetilde{\mathcal{M}} = (\mathcal{S}, \mathcal{A}, P, \widetilde{r}, \widetilde{\gamma})$ for a fixed $\lambda \in [0, 1]$, where $\widetilde{r}, \widetilde{\gamma}$ are defined in (1). We can view this MDP as the one in a single iteration of HuRL. For a policy $\pi$, we denote its state value function in $\widetilde{\mathcal{M}}$ as $\widetilde{V}^\pi$, and the optimal policy and value function of $\widetilde{\mathcal{M}}$ as $\widetilde{\pi}^*$ and $\widetilde{V}^*$, respectively. The missing proofs of the results from this section can be found in Appendix A.

### 4.1 Short-Horizon Reduction: Performance Decomposition

Our main result is a performance decomposition, which characterizes how a heuristic $h$ and suboptimality in solving the reshaped MDP $\widetilde{\mathcal{M}}$ relate to performance in the original MDP $\mathcal{M}$.

**Theorem 4.1.** *For any policy $\pi$, heuristic $f : \mathcal{S} \to \mathbb{R}$, and mixing coefficient $\lambda \in [0, 1]$,*

$$V^*(d_0) - V^\pi(d_0) = \text{Regret}(h, \lambda, \pi) + \text{Bias}(h, \lambda, \pi)$$

*where we define*

$$\text{Regret}(h, \lambda, \pi) \coloneqq \lambda \left( \widetilde{V}^*(d_0) - \widetilde{V}^\pi(d_0) \right) + \frac{1-\lambda}{1-\gamma} \left( \widetilde{V}^*(d^\pi) - \widetilde{V}^\pi(d^\pi) \right) \tag{3}$$

$$\text{Bias}(h, \lambda, \pi) \coloneqq \left( V^*(d_0) - \widetilde{V}^*(d_0) \right) + \frac{\gamma(1-\lambda)}{1-\gamma} \mathbb{E}_{s,a \sim d^\pi} \mathbb{E}_{s'|s,a} \left[ h(s') - \widetilde{V}^*(s') \right] \tag{4}$$

*Furthermore, $\forall b \in \mathbb{R}$, $\text{Bias}(h, \lambda, \pi) = \text{Bias}(h + b, \lambda, \pi)$ and $\text{Regret}(h, \lambda, \pi) = \text{Regret}(h + b, \lambda, \pi)$.*

The theorem shows that suboptimality of a policy $\pi$ in the original MDP $\mathcal{M}$ can be decomposed into *1)* a *bias* term due to solving a reshaped MDP $\widetilde{\mathcal{M}}$ instead of the original MDP $\mathcal{M}$, and *2)* a *regret* term (i.e. the learning variance) due to $\pi$ being suboptimal in the reshaped MDP $\widetilde{\mathcal{M}}$. Moreover, it shows that heuristics are equivalent up to constant offsets. In other words, only the relative ordering between states that a heuristic induces matters in learning, not the absolute values.

Balancing the two terms trades off bias and variance in learning. Using a smaller $\lambda$ replaces the long-term information with the heuristic and make the horizon of the reshaped MDP $\widetilde{\mathcal{M}}$ shorter. Therefore, given a finite interaction budget, the regret term in (3) can be more easily minimized, though the bias term in (4) can potentially be large if the heuristic is bad. On the contrary, with $\lambda = 1$, the bias is completely removed, as the agent solves the original MDP $\mathcal{M}$ directly.

## 4.2 Regret, Algorithm Requirement, and Relationship with PBRS

The regret term in (3) characterizes the performance gap due to $\pi$ being suboptimal in the reshaped MDP $\widetilde{\mathcal{M}}$, because $\mathrm{Regret}(h, \lambda, \widetilde{\pi}^*) = 0$ for any $h$ and $\lambda$. For learning, we need the base RL algorithm $\mathcal{L}$ to find a policy $\pi$ such that the regret term in (3) is small. By the definition in (3), the base RL algorithm $\mathcal{L}$ is required not only to find a policy $\pi$ such that $\widetilde{V}^*(s) - \widetilde{V}^\pi(s)$ is small for states from $d_0$, *but also for states $\pi$ visits when rolling out in the original MDP $\mathcal{M}$*. In other words, it is insufficient for the base RL algorithm to only optimize for $\widetilde{V}^\pi(d_0)$ (the performance in the reshaped MDP with respect to the initial state distribution; see Section 2.2). For example, suppose $\lambda = 0$ and $d_0$ concentrates on a single state $s_0$. Then maximizing $\widetilde{V}^\pi(d_0)$ alone would only optimize $\pi(\cdot|s_0)$ and the policy $\pi$ need not know how to act in other parts of the state space.

To use HuRL, we need the base algorithm to learn a policy $\pi$ that has small *action gaps* in the reshaped MDP $\widetilde{\mathcal{M}}$ *but along trajectories in the original MDP $\mathcal{M}$*, as we show below. This property is satisfied by off-policy RL algorithms such as Q-learning [34].

**Proposition 4.1.** *For any policy $\pi$, heuristic $f : \mathcal{S} \to \mathbb{R}$ and mixing coefficient $\lambda \in [0, 1]$,*

$$\mathrm{Regret}(h, \lambda, \pi) = -\mathbb{E}_{\rho^\pi(d_0)}\left[\sum_{t=0}^\infty \gamma^t \widetilde{A}^*(s_t, a_t)\right]$$

*where $\rho^\pi(d_0)$ denotes the trajectory distribution of running $\pi$ from $d_0$, and $\widetilde{A}^*(s, a) = \widetilde{r}(s, a) + \widetilde{\gamma}\mathbb{E}_{s'|s,a}[\widetilde{V}^*(s')] - \widetilde{V}^*(s) \leq 0$ is the action gap with respect to the optimal policy $\widetilde{\pi}^*$ of $\widetilde{\mathcal{M}}$.*

Another way to comprehend the regret term is through studying its dependency on $\lambda$. When $\lambda = 1$, $\mathrm{Regret}(h, 0, \pi) = V^*(d_0) - V^\pi(d_0)$, which is identical to the policy regret in $\mathcal{M}$ for a *fixed* initial distribution $d_0$. On the other hand, when $\lambda = 0$, $\mathrm{Regret}(h, 0, \pi) = \max_{\pi'} \frac{1}{1-\gamma}\mathbb{E}_{s\sim d^\pi}[\widetilde{r}(s, \pi') - \widetilde{r}(s, \pi)]$, which is the regret of a *non-stationary* contextual bandit problem where the context distribution is $d^\pi$ (the average state distribution of $\pi$). In general, for $\lambda \in (0, 1)$, the regret notion mixes a short-horizon non-stationary problem and a long-horizon stationary problem.

One natural question is whether the reshaped MDP $\widetilde{\mathcal{M}}$ has a more complicated and larger value landscape than the original MDP $\mathcal{M}$, because these characteristics may affect the regret rate of a base algorithm. We show that $\widetilde{\mathcal{M}}$ preserves the value bounds and linearity of the original MDP.

**Proposition 4.2.** *Reshaping the MDP as in* (1) *preserves the following characteristics: 1) If $h(s) \in [0, \frac{1}{1-\gamma}]$, then $\widetilde{V}^\pi(s) \in [0, \frac{1}{1-\gamma}]$ for all $\pi$ and $s \in \mathcal{S}$. 2) If $\widetilde{\mathcal{M}}$ is a linear MDP with feature vector $\phi(s, a)$ (i.e. $r(s, a)$ and $\mathbb{E}_{s'|s,a}[g(s')]$ for any $g$ can be linearly parametrized in $\phi(s, a)$), then $\widetilde{\mathcal{M}}$ is also a linear MDP with feature vector $\phi(s, a)$.*

On the contrary, the MDP $\overline{\mathcal{M}}_\lambda := (\mathcal{S}, \mathcal{A}, P, \overline{r}, \lambda\gamma)$ in Section 2.2 does not have these properties. We can show that $\overline{\mathcal{M}}_\lambda$ is equivalent to $\widetilde{\mathcal{M}}$ up to a PBRS transformation (i.e., $\overline{r}(s, a) = \widetilde{r}(s, a) + \widetilde{\gamma}\mathbb{E}_{s'|s,a}[h(s')] - h(s)$). Thus, HuRL incorporates guidance discount into PBRS with nicer properties.

## 4.3 Bias and Heuristic Quality

The bias term in (4) characterizes suboptimality due to using a heuristic $h$ in place of long-term state values in $\mathcal{M}$. What is the best heuristic in this case? From the definition of the bias term in (4), we see that the ideal heuristic is the optimal value $V^*$, as $\mathrm{Bias}(V^*, \lambda, \pi) = 0$ for all $\lambda \in [0, 1]$. By continuity, we can expect that if $h$ deviates from $V^*$ a little, then the bias is small.

**Corollary 4.1.** *If $\inf_{b\in\mathbb{R}} \|h + b - V^*\|_\infty \leq \epsilon$, then $\mathrm{Bias}(h, \lambda, \pi) \leq \frac{(1-\lambda\gamma)^2}{(1-\gamma)^2}\epsilon$.*

To better understand how the heuristic $h$ affects the bias, we derive an upper bound on the bias by replacing the first term in (4) with an upper bound that depends only on $\pi^*$.

**Proposition 4.3.** *For $g : \mathcal{S} \to \mathbb{R}$ and $\eta \in [0,1]$, define $\mathcal{C}(\pi, g, \eta) := \mathbb{E}_{\rho^\pi(d_0)}\left[\sum_{t=1}^\infty \eta^{t-1} g(s_t)\right]$. Then $\mathrm{Bias}(h, \lambda, \pi) \le (1-\lambda)\gamma(\mathcal{C}(\pi^*, V^* - h, \lambda\gamma) + \mathcal{C}(\pi, h - \widetilde{V}^*, \gamma))$.*

In Proposition 4.3, the term $(1-\lambda)\gamma\mathcal{C}(\pi^*, V^* - h, \lambda\gamma)$ is the underestimation error of the heuristic $h$ under the states visited by the optimal policy $\pi^*$ in the original MDP $\mathcal{M}$. Therefore, to minimize the first term in the bias, we would want the heuristic $h$ to be large along the paths that $\pi^*$ generates.

However, Proposition 4.3 also discourages the heuristic from being arbitrarily large, because the second term in the bias in (4) (or, equivalently, the second term in Proposition 4.3) incentivizes the heuristic to underestimate the optimal value of the reshaped MDP $\widetilde{V}^*$. More precisely, the second term requires the heuristic to obey some form of spatial consistency. A quick intuition is the observation that if $h(s) = V^{\pi'}(s)$ for some $\pi'$ or $h(s) = 0$, then $h(s) \le \widetilde{V}^*(s)$ for all $s \in \mathcal{S}$. More generally, we show that if the heuristic is *improvable* with respect to the original MDP $\mathcal{M}$ (i.e. the heuristic value is lower than that of the max of Bellman backup), then $h(s) \le \widetilde{V}^*(s)$. By Proposition 4.3, learning with an improvable heuristic in HuRL has a much smaller bias.

**Definition 4.1.** Define the Bellman operator $(\mathcal{B}h)(s,a) := r(s,a) + \gamma\mathbb{E}_{s'|s,a}[h(s')]$. A heuristic function $h : \mathcal{S} \to \mathbb{R}$ is said to be *improvable* with respect to an MDP $\mathcal{M}$ if $\max_a(\mathcal{B}h)(s,a) \ge h(s)$.

**Proposition 4.4.** *If $h$ is improvable with respect to $\mathcal{M}$, then $\widetilde{V}^*(s) \ge h(s)$, for all $\lambda \in [0,1]$.*

### 4.4 Pessimistic Heuristics are Good Heuristics

While Corollary 4.1 shows that HuRL can handle an imperfect heuristic, this result is not ideal. The corollary depends on the $\ell_\infty$ approximation error, which can be difficult to control in large state spaces. Here we provide a more refined sufficient condition of good heuristics. We show that the concept of *pessimism* in the face of uncertainty provides a finer mechanism for controlling the approximation error of a heuristic and would allow us to remove the $\ell_\infty$-type error. This result is useful for constructing heuristics from data that does not have sufficient support.

From Proposition 4.3 we see that the source of the $\ell_\infty$ error is the second term in the bias upper bound, as it depends on the states that the agent's policy visits which can change during learning. To remove this dependency, we can use improvable heuristics (see Proposition 4.4), as they satisfy $h(s) \le \widetilde{V}^*(s)$. Below we show that Bellman-consistent pessimism yields improvable heuristics.

**Proposition 4.5.** *Suppose $h(s) = Q(s, \pi')$ for some policy $\pi'$ and function $Q : \mathcal{S} \times \mathcal{A} \to \mathbb{R}$ such that $Q(s,a) \le (\mathcal{B}h)(s,a)$, $\forall s \in \mathcal{S}$, $a \in \mathcal{A}$. Then $h$ is improvable and $f(s) \le V^{\pi'}(s)$ for all $s \in \mathcal{S}$.*

The Bellman-consistent pessimism in Proposition 4.5 essentially says that $h$ is pessimistic with respect to the Bellman backup. This condition has been used as the foundation for designing pessimistic off-policy RL algorithms, such as pessimistic value iteration [30] and algorithms based on pessimistic absorbing MDPs [31]. In other words, these pessimistic algorithms can be used to construct good heuristics with small bias in Proposition 4.3 from offline data. With such a heuristic, the bias upper bound would be simply $\mathrm{Bias}(h, \lambda, \pi) \le (1-\lambda)\gamma\mathcal{C}(\pi^*, V^* - h, \lambda\gamma)$. Therefore, as long as enough batch data are sampled from a distribution that covers states that $\pi^*$ visits, these pessimistic algorithms can construct good heuristics with nearly zero bias for HuRL with high probability.

## 5   Experiments

We validate our framework HuRL experimentally in MuJoCo (commercial license) [32] robotics control problems and Procgen games (MIT License) [33], where soft actor critic (SAC) [35] and proximal policy optimization (PPO) [36] were used as the base RL algorithms, respectively[5]. The goal is to study whether HuRL can accelerate learning by shortening the horizon with heuristics. In particular, we conduct studies to investigate the effects of different heuristics and mixing coefficients. Since the main focus here is on the possibility of leveraging a *given* heuristic to accelerate RL algorithms, in these experiments we used vanilla techniques to construct heuristics for HuRL. Experimentally studying the design of heuristics for a domain or a batch of data is beyond the scope of the current paper but are important future research directions. For space limitation, here we report only the results of the MuJoCo experiments. The results on Procgen games along with other experimental details can also be found in Appendix C.

---

[5]Code to replicate all experiments is available at https://github.com/microsoft/HuRL.

### 5.1 Setup

We consider four MuJoCo environments with dense rewards (Hopper-v2, HalfCheetah-v2, Humanoid-v2, and Swimmer-v2) and a sparse reward version of Reacher-v2 (denoted as Sparse-Reacher-v2)[6]. We design the experiments to simulate two learning scenarios. First, we use Sparse-Reacher-v2 to simulate the setting where an engineered heuristic based on domain knowledge is available; since this is a goal reaching task, we designed a heuristic $h(s) = r(s, a) - 100\|e(s) - g(s)\|$, where $e(s)$ and $g(s)$ denote the robot's end-effector position and the goal position, respectively. Second, we use the dense reward environments to model scenarios where a batch of data collected by multiple behavioral policies is available before learning, and a heuristic is constructed by an offline policy evaluation algorithm from the batch data (see Appendix C.1 for details). In brief, we generated these behavioral policies by running SAC from scratch and saved the intermediate policies generated in training. We then use least-squares regression to fit a neural network to predict empirical Monte-Carlo returns of the trajectories in the sampled batch of data. We also use behavior cloning (BC) to warm-start all RL agents based on the same batch dataset in the dense reward experiments.

The base RL algorithm here, SAC, is based on the standard implementation in Garage (MIT License) [37]. The policy and value networks are fully connected independent neural networks. The policy is Tanh-Gaussian and the value network has a linear head.

**Algorithms.** We compare the performance of different algorithms below. *1)* BC *2)* SAC *3)* SAC with BC warm start (SAC w/ BC) *4)* HuRL with the engineered heuristic (HuRL) *5)* HuRL with a zero heuristic and BC warm start (HuRL-zero) *6)* HuRL with the Monte-Carlo heuristic and BC warm start (HuRL-MC) *7)* SAC with PBRS reward (and BC warm start, if applicable) (PBRS). For the HuRL algorithms, the mixing coefficient was scheduled as $\lambda_n = \lambda_0 + (1 - \lambda_0)c_\omega \tanh(\omega(n - 1))$, for $n = 1, \ldots, N$, where $\lambda_0 \in [0, 1]$, $\omega > 0$ controls the increasing rate, and $c_\omega$ is a normalization constant such that $\lambda_\infty = 1$ and $\lambda_n \in [0, 1]$. We chose these algorithms to study the effect of each additional warm-start component (BC and heuristics) added on top of vanilla SAC. HuRL-zero is SAC w/ BC but with an extra $\lambda$ schedule above that further lowers the discount, whereas SAC and SAC w/ BC keep a constant discount factor.

**Evaluation and Hyperparameters.** In each iteration, the RL agent has a fixed sample budget for environment interactions, and its performance is measured in terms of undiscounted cumulative returns of the deterministic mean policy extracted from SAC. The hyperparameters used in the algorithms above were selected as follows. First, the learning rates and the discount factor of the base RL algorithm, SAC, were tuned for each environment. The tuned discount factor was used as the discount factor $\gamma$ of the original MDP $\mathcal{M}$. Fixing the hyperparameters above, we additionally tune $\lambda_0$ and $\omega$ for the $\lambda$ schedule of HuRL for each environment and each heuristic. Finally, after all these hyperparameters were fixed, we conducted additional testing runs with 30 different random seeds and report their statistics here. Sources of randomness included the data collection process of the behavioral policies, training the heuristics from batch data, BC, and online RL. However, the behavioral policies were fixed across all testing runs. We chose this hyperparameter tuning procedure to make sure that the baselines (i.e. SAC) compared in these experiments are their best versions.

### 5.2 Results Summary

Fig. 2 shows the results on the MuJoCo environments. Overall, we see that HuRL is able to leverage engineered and learned heuristics to significantly improve the learning efficiency. This trend is consistent across all environments that we tested on.

For the sparse-reward experiments, we see that SAC and PBRS struggle to learn, while HuRL is able to converge to the optimal performance much faster. For the dense reward experiments, similarly HuRL-MC converges much faster, though the gain in HalfCheetah-v2 is minor and it might have converged to a worse local maximum in Swimmer-v2. In addition, we see that warm-starting SAC using BC (i.e. SAC w/ BC) can improve the learning efficiency compared with the vanilla SAC, but using BC alone does not result in a good policy. Lastly, we see that using the zero heuristic (HuRL-zero) with extra $\lambda$-scheduling does not further improve the performance of SAC w/ BC. This comparison verifies that the learned Monte-Carlo heuristic provides non-trivial information.

Interestingly, we see that applying PBRS to SAC leads to even worse performance than running SAC with the original reward. There are two reasons why SAC+PBRS is less desirable than SAC+HuRL

---

[6]The reward is zero at the goal and $-1$ otherwise.

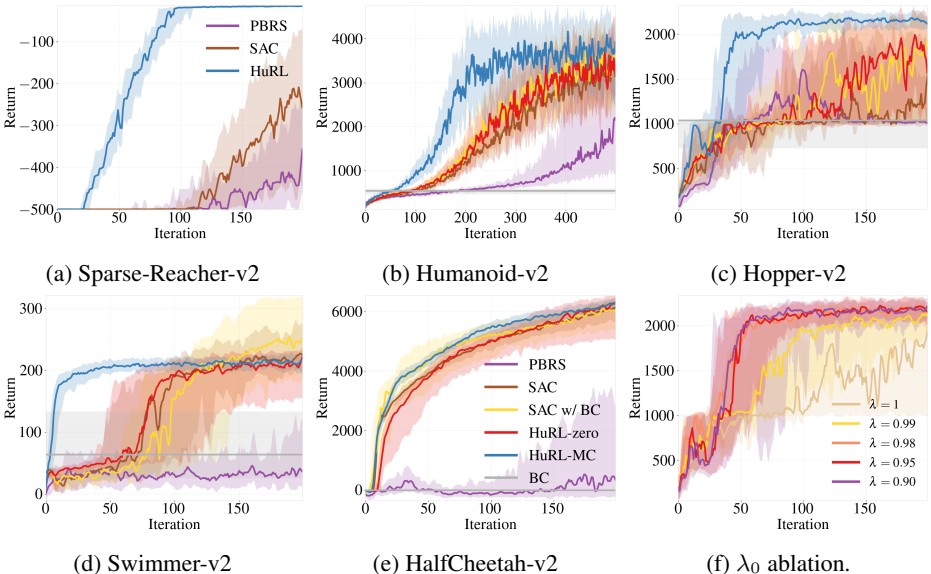

| (a) Sparse-Reacher-v2 | (b) Humanoid-v2 | (c) Hopper-v2 |
| (d) Swimmer-v2 | (e) HalfCheetah-v2 | (f) $\lambda_0$ ablation. |

Figure 2: Experimental results. (a) uses an engineered heuristic for a sparse reward problem; (b)-(e) use heuristics learned from offline data and share the same legend; (e) shows ablation results of different initial $\lambda_0$ in Hopper-v2. The plots show the 25th, 50th, 75th percentiles of algorithm performance over 30 random seeds.

as we discussed before: *1)* PBRS changes the reward/value scales in the induced MDP, and popular RL algorithms like SAC are very sensitive to such changes. In contrast, HuRL induces values on the same scale as we show in Proposition 4.2. *2)* In HuRL, we are effectively providing the algorithm some more side-information to let SAC shorten the horizon when the heuristic is good.

The results in Fig. 2 also have another notable aspect. Because the datasets used in the dense reward experiments contain trajectories collected by a range of policies, it is likely that BC suffers from disagreement in action selection among different policies. Nonetheless, training a heuristic using a basic Monte-Carlo regression seems to be less sensitive to these conflicts and still results in a helpful heuristic for HuRL. One explanation can be that heuristics are only functions of states, not of states and actions, and therefore the conflicts are minor. Another plausible explanation is that HuRL only uses the heuristic to *guide* learning, and does not completely rely on it to make decisions Thus, HuRL can be more robust to the heuristic quality, or, equivalently, to the quality of prior knowledge.

### 5.3 Ablation: Effects of Horizon Shortening

To further verify that the acceleration in Fig. 2 is indeed due to horizon shortening, we conducted an ablation study for HuRL-MC on Hopper-v2, whose results are presented in Fig. 2f. HuRL-MC's best $\lambda$-schedule hyperparameters on Hopper-v2, which are reflected in its performance in the aforementioned Fig. 2c, induced a near-constant schedule at $\lambda = 0.95$; to obtain the curves in Fig. 2f, we ran HuRL-MC with constant-$\lambda$ schedules for several more $\lambda$ values. Fig. 2f shows that increasing $\lambda$ above $0.98$ leads to a performance drop. Since using a large $\lambda$ decreases bias and makes the reshaped MDP more similar to the original MDP, we conclude that the increased learning speed on Hopper-v2 is due to HuRL's horizon shortening (coupled with the guidance provided by its heuristic).

## 6 Related Work

**Discount regularization.** The horizon-truncation idea can be traced back to Blackwell optimality in the known MDP setting [18]. Reducing the discount factor amounts to running HuRL with a zero heuristic. Petrik and Scherrer [19], Jiang et al. [20, 21] study the MDP setting; Chen et al. [22] study POMDPs. Amit et al. [23] focus on discount regularization for Temporal Difference (TD) methods, while Van Seijen et al. [6] use a logarithmic mapping to lower the discount for online RL.

**Reward shaping.** Reward shaping has a long history in RL, from the seminal PBRS work [29] to recent bilevel-optimization approaches [38]. Tessler and Mannor [5] consider a complementary problem to HuRL: given a discount $\gamma'$, they find a reward $r'$ that preserves trajectory ordering in the original MDP. Meanwhile there is a vast literature on bias-variance trade-off for online RL with

horizon truncation. TD($\lambda$) [39, 40] and Generalized Advantage Estimates [41] blend value estimates across discount factors, while Sherstan et al. [42] use the discount factor as an input to the value function estimator. TD($\Delta$) [43] computes differences between value functions across discount factors.

**Heuristics in model-based methods.** Classic uses of heuristics include A* [24], Monte-Carlo Tree Search (MCTS) [25], and Model Predictive Control (MPC) [44]. Zhong et al. [26] shorten the horizon in MPC using a value function approximator. Hoeller et al. [27] additionally use an estimate for the running cost to trade off solution quality and amount of computation. Bejjani et al. [28] show heuristic-accelerated truncated-horizon MPC on actual robots and tune the value function throughout learning. Bhardwaj et al. [7] augment MPC with a terminal value heuristic, which can be viewed as an instance of HuRL where the base algorithm is MPC. Asai and Muise [45] learn an MDP expressible in the STRIPS formalism that can benefit from relaxation-based planning heuristics. But HuRL is more general, as it does not assume model knowledge and can work in unknown environments.

**Pessimistic extrapolation.** Offline RL techniques employ pessimistic extrapolation for robustness [30], and their learned value functions can be used as heuristics in HuRL. Kumar et al. [46] penalize out-of-distribution actions in off-policy optimization while Liu et al. [31] additionally use a variational auto-encoder (VAE) to detect out-of-distribution states. We experimented with VAE-filtered pessimistic heuristics in Appendix C. Even pessimistic offline evaluation techniques [16] can be useful in HuRL, since function approximation often induces extrapolation errors [47].

**Heuristic pessimism vs. admissibility.** Our concept of heuristic pessimism can be easily confused for the well-established notion of *admissibility* [48], but in fact they are opposites. Namely, an admissible heuristic never *underestimates* $V^*$ (in the return-maximization setting), while a pessimistic one never *overestimates* $V^*$. Similarly, our notion of improvability is distinct from *consistency*: they express related ideas, but with regards to pessimistic and admissible value functions, respectively. Thus, counter-intuitively from the planning perspective, our work shows that for policy *learning*, *in*admissible heuristics are desirable. Pearl [49] is one of the few works that has analyzed desirable implications of heuristic inadmissibility in planning.

**Other warm-starting techniques.** HuRL is a new way to warm-start online RL methods. Bianchi et al. [50] use a heuristic policy to initialize agents' policies. Vinyals et al. [2], Hester et al. [10] train a value function and policy using batch IL and then used them as regularization in online RL. Nair et al. [9] use off-policy RL on batch data and fine-tune the learned policy. Recent approaches of hybrid IL-RL have strong connections to HuRL [17, 51, 52]. In particular, Cheng et al. [17] is a special case of HuRL with a max-aggregation heuristic. Farahmand et al. [8] use several related tasks to learn a task-dependent heuristic and perform shorter-horizon planning or RL. Knowledge distillation approaches [53] can also be used to warm-start learning, but in contrast to them, HuRL expects prior knowledge in the form of state value estimates, not features, and doesn't attempt to make the agent internalize this knowledge. A HuRL agent learns from its own environment interactions, using prior knowledge only as guidance. Reverse Curriculum approaches [54] create short horizon RL problems by initializing the agent close to the goal, and moving it further away as the agent improves. This gradual increase in the horizon inspires the HuRL approach. However, HuRL does not require the agent to be initialized on expert states and can work with many different base RL algorithms.

## 7   Discussion and Limitations

This work is an early step towards theoretically understanding the role and potential of heuristics in guiding RL algorithms. We propose a framework, HuRL, that can accelerate RL when an informative heuristic is provided. HuRL induces a horizon-based regularization of RL, complementary to existing warm-starting schemes, and we provide theoretical and empirical analyses to support its effectiveness. While this is a conceptual work without foreseeable societal impacts yet, we hope that it will help counter some of AI's risks by making RL more predictable via incorporating prior into learning.

We remark nonetheless that the effectiveness of HuRL depends on the available heuristic. While HuRL can eventually solve the original RL problem even with a non-ideal heuristic, using a bad heuristic can slow down learning. Therefore, an important future research direction is to adaptively tune the mixing coefficient based on the heuristic quality with curriculum or meta-learning techniques. In addition, while our theoretical analysis points out a strong connection between good heuristics for HuRL and pessimistic offline RL, techniques for the latter are not yet scalable and robust enough for high-dimensional problems. Further research on offline RL can unlock the full potential of HuRL.

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
