# A Missing Proofs

We provide the complete proofs of the theorems stated in the main paper. We defer the proofs of the technical results to Appendix B.

## A.1 Proof of Theorem 4.1

**Theorem 4.1.** *For any policy $\pi$, heuristic $f : \mathcal{S} \to \mathbb{R}$, and mixing coefficient $\lambda \in [0, 1]$,*

$$V^*(d_0) - V^\pi(d_0) = \mathrm{Regret}(h, \lambda, \pi) + \mathrm{Bias}(h, \lambda, \pi)$$

*where we define*

$$\mathrm{Regret}(h, \lambda, \pi) := \lambda \left( \widetilde{V}^*(d_0) - \widetilde{V}^\pi(d_0) \right) + \frac{1 - \lambda}{1 - \gamma} \left( \widetilde{V}^*(d^\pi) - \widetilde{V}^\pi(d^\pi) \right) \tag{3}$$

$$\mathrm{Bias}(h, \lambda, \pi) := \left( V^*(d_0) - \widetilde{V}^*(d_0) \right) + \frac{\gamma(1 - \lambda)}{1 - \gamma} \mathbb{E}_{s,a \sim d^\pi} \mathbb{E}_{s'|s,a} \left[ h(s') - \widetilde{V}^*(s') \right] \tag{4}$$

*Furthermore, $\forall b \in \mathbb{R}$, $\mathrm{Bias}(h, \lambda, \pi) = \mathrm{Bias}(h + b, \lambda, \pi)$ and $\mathrm{Regret}(h, \lambda, \pi) = \mathrm{Regret}(h + b, \lambda, \pi)$.*

First we prove the equality using a new performance difference lemma that we will prove in Appendix B. This result may be of independent interest.

**Lemma A.1** (General Performance Difference Lemma). *Consider the reshaped MDP $\widetilde{\mathcal{M}}$ defined by some $f : \mathcal{S} \to \mathbb{R}$ and $\lambda \in [0, 1]$. For any policy $\pi$, any state distribution $d_0$ and any $V : \mathcal{S} \to \mathbb{R}$, it holds that*

$$V(d_0) - V^\pi(d_0) = \frac{\gamma(1 - \lambda)}{1 - \gamma} \mathbb{E}_{s,a \sim d^\pi} \mathbb{E}_{s'|s,a} \left[ h(s') - V(s') \right]$$
$$+ \lambda \left( V(d_0) - \widetilde{V}^\pi(d_0) \right) + \frac{1 - \lambda}{1 - \gamma} \left( V(d^\pi) - \widetilde{V}^\pi(d^\pi) \right)$$

Now take $V$ as $\widetilde{V}^*$ in the equality above. Then we can write

$$V^*(d_0) - V^\pi(d_0) = \left( V^*(d_0) - \widetilde{V}^*(d_0) \right) + \frac{\gamma(1 - \lambda)}{1 - \gamma} \mathbb{E}_{s,a \sim d^\pi} \mathbb{E}_{s'|s,a} \left[ h(s') - \widetilde{V}^*(s') \right]$$
$$+ \lambda \left( \widetilde{V}^*(d_0) - \widetilde{V}^\pi(d_0) \right) + \frac{1 - \lambda}{1 - \gamma} \left( \widetilde{V}^*(d^\pi) - \widetilde{V}^\pi(d^\pi) \right)$$

which is the regret-bias decomposition.

Next we prove that these two terms are independent of constant offsets. For the regret term, this is obvious because shifting the heuristic by a constant would merely shift the reward by a constant. For the bias term, we prove the invariance below.

**Proposition A.1.** $\mathrm{Bias}(h, \lambda, \pi) = \mathrm{Bias}(h + b, \lambda, \pi)$ *for any $b \in \mathbb{R}$.*

*Proof.* Notice that any $b \in \mathbb{R}$ and $\pi$, $\widetilde{V}^\pi(s; f + b) - \widetilde{V}^\pi(s; f) = \sum_{t=0}^\infty (\lambda\gamma)^t (1 - \lambda)\gamma b = \frac{(1-\lambda)\gamma}{1-\lambda\gamma} b$. Therefore, we can derive

$$\mathrm{Bias}(h + b, \lambda, \pi) - \mathrm{Bias}(h, \lambda, \pi) = -\frac{(1 - \lambda)\gamma}{1 - \gamma\lambda} b + \frac{\gamma(1 - \lambda)}{1 - \gamma} \mathbb{E}_{s,a \sim d^\pi} \mathbb{E}_{s'|s,a} \left[ b - \frac{(1 - \lambda)\gamma}{1 - \gamma\lambda} b \right]$$
$$= \frac{\gamma(1 - \lambda)}{1 - \gamma} b - \left( 1 + \frac{\gamma(1 - \lambda)}{1 - \gamma} \right) \frac{(1 - \lambda)\gamma}{1 - \gamma\lambda} b$$

Since

$$\left( 1 + \frac{\gamma(1 - \lambda)}{1 - \gamma} \right) \frac{(1 - \lambda)\gamma}{1 - \gamma\lambda} b = \frac{1 - \gamma + \gamma(1 - \lambda)}{1 - \gamma} \frac{(1 - \lambda)\gamma}{1 - \gamma\lambda} b = \frac{1 - \gamma\lambda}{1 - \gamma} \frac{(1 - \lambda)\gamma}{1 - \gamma\lambda} b = \frac{(1 - \lambda)\gamma}{1 - \gamma} b$$

we have $\mathrm{Bias}(h + b, \lambda, \pi) - \mathrm{Bias}(h, \lambda, \pi) = 0$. $\blacksquare$

## A.2 Proof of Proposition 4.1

**Proposition 4.1.** *For any policy $\pi$, heuristic $f : \mathcal{S} \to \mathbb{R}$ and mixing coefficient $\lambda \in [0,1]$,*

$$\text{Regret}(h, \lambda, \pi) = -\mathbb{E}_{\rho^\pi(d_0)}\left[\sum_{t=0}^\infty \gamma^t \widetilde{A}^*(s_t, a_t)\right]$$

*where $\rho^\pi(d_0)$ denotes the trajectory distribution of running $\pi$ from $d_0$, and $\widetilde{A}^*(s, a) = \widetilde{r}(s, a) + \widetilde{\gamma}\mathbb{E}_{s'|s,a}[\widetilde{V}^*(s')] - \widetilde{V}^*(s) \le 0$ is the action gap with respect to the optimal policy $\widetilde{\pi}^*$ of $\widetilde{\mathcal{M}}$.*

Define the Bellman backup for the reshaped MDP:

$$(\widetilde{\mathcal{B}}V)(s, a) := \widetilde{r}(s, a) + \widetilde{\gamma}\mathbb{E}_{s'|s,a}[V(s')]$$

Then by Lemma B.6 in Appendix B, we can rewrite the regret as

$$\lambda\left(\widetilde{V}^*(d_0) - \widetilde{V}^\pi(d_0)\right) + \frac{1-\lambda}{1-\gamma}\left(\widetilde{V}^*(d^\pi) - \widetilde{V}^\pi(d^\pi)\right) = \mathbb{E}_{\rho^\pi(d_0)}\left[\sum_{t=0}^\infty \gamma^t\left(\widetilde{V}^*(s_t) - (\widetilde{\mathcal{B}}\widetilde{V}^*)(s_t, a_t)\right)\right]$$

Notice the equivalence $\widetilde{V}^*(s) - (\widetilde{\mathcal{B}}\widetilde{V}^*)(s, a) = -\widetilde{A}^*(s, a)$. This concludes the proof.

## A.3 Proof of Proposition 4.2

**Proposition 4.2.** *Reshaping the MDP as in* (1) *preserves the following characteristics: 1) If $h(s) \in [0, \frac{1}{1-\gamma}]$, then $\widetilde{V}^\pi(s) \in [0, \frac{1}{1-\gamma}]$ for all $\pi$ and $s \in \mathcal{S}$. 2) If $\widetilde{\mathcal{M}}$ is a linear MDP with feature vector $\phi(s, a)$ (i.e. $r(s, a)$ and $\mathbb{E}_{s'|s,a}[g(s')]$ for any $g$ can be linearly parametrized in $\phi(s, a)$), then $\widetilde{\mathcal{M}}$ is also a linear MDP with feature vector $\phi(s, a)$.*

For the first statement, notice $\widetilde{r}(s, a) \in [0, 1 + \frac{(1-\lambda)\gamma}{1-\gamma}]$. Therefore, we have $\widetilde{V}^\pi(s) \ge 0$ as well as

$$\widetilde{V}^\pi(s) \le \frac{1}{1-\lambda\gamma}\left(1 + \frac{(1-\lambda)\gamma}{1-\gamma}\right)$$

$$= \frac{1}{1-\lambda\gamma}\frac{1-\gamma+(1-\lambda)\gamma}{1-\gamma} = \frac{1}{1-\gamma}$$

For the second statement, we just need to show the reshaped reward $\widetilde{r}(s, a)$ is linear in $\phi(s, a)$. This is straightforward because $\mathbb{E}_{s'|s,a}[h(s')]$ is linear in $\phi(s, a)$.

## A.4 Proof of Corollary 4.1

**Corollary 4.1.** *If $\inf_{b\in\mathbb{R}} \|h + b - V^*\|_\infty \le \epsilon$, then $\text{Bias}(h, \lambda, \pi) \le \frac{(1-\lambda\gamma)^2}{(1-\gamma)^2}\epsilon$.*

By Theorem 4.1, we know that $\text{Bias}(h, \lambda, \pi) = \text{Bias}(h+b, \lambda, \pi)$ for any $b \in \mathbb{R}$. Now consider $b^* \in \mathbb{R}$ such that $\|h + b^* - V^*\|_\infty \le \epsilon$. Then by Lemma B.5, we have also $\|h + b^* - \widetilde{V}^{\pi^*}\|_\infty \le \epsilon + \frac{(1-\lambda)\gamma\epsilon}{1-\lambda\gamma}$.

Therefore, by Proposition 4.3, we can derive with definition of the bias,

$$\begin{aligned}
\text{Bias}(h, \lambda, \pi) &= \text{Bias}(h + b^*, \lambda, \pi) \\
&\le (1-\lambda)\gamma\left(\mathcal{C}(\pi^*, V^* - h - b^*, \lambda\gamma) + \mathcal{C}(\pi, h + b^* - \widetilde{V}^*, \gamma)\right) \\
&\le (1-\lambda)\gamma\left(\mathcal{C}(\pi^*, V^* - h - b^*, \lambda\gamma) + \mathcal{C}(\pi, h + b^* - \widetilde{V}^{\pi^*}, \gamma)\right) \\
&\le (1-\lambda)\gamma\left(\frac{\epsilon}{1-\lambda\gamma} + \frac{1}{1-\gamma}(\epsilon + \frac{(1-\lambda)\gamma\epsilon}{1-\lambda\gamma})\right) \\
&\le (1-\lambda)\gamma\left(\frac{\epsilon}{1-\gamma} + \frac{1}{1-\gamma}(\epsilon + \frac{(1-\lambda)\gamma\epsilon}{1-\gamma})\right) \\
&= \frac{2(1-\lambda)\gamma\epsilon}{1-\gamma} + \frac{(1-\lambda)^2\gamma^2\epsilon}{(1-\gamma)^2} \le \frac{(1-\lambda\gamma)^2}{(1-\gamma)^2}\epsilon
\end{aligned}$$

## A.5 Proof of Proposition 4.3

**Proposition 4.3.** *For $g : \mathcal{S} \to \mathbb{R}$ and $\eta \in [0, 1]$, define $\mathcal{C}(\pi, g, \eta) := \mathbb{E}_{\rho^\pi(d_0)} \left[ \sum_{t=1}^\infty \eta^{t-1} g(s_t) \right]$. Then $\mathrm{Bias}(h, \lambda, \pi) \le (1 - \lambda)\gamma(\mathcal{C}(\pi^*, V^* - h, \lambda\gamma) + \mathcal{C}(\pi, h - \widetilde{V}^*, \gamma))$.*

Recall the definition of bias:

$$\mathrm{Bias}(h, \lambda, \pi) = \left( V^*(d_0) - \widetilde{V}^*(d_0) \right) + \frac{\gamma(1 - \lambda)}{1 - \gamma} \mathbb{E}_{s,a \sim d^\pi} \mathbb{E}_{s'|s,a} \left[ h(s') - \widetilde{V}^*(s') \right]$$

For the first term, we can derive by performance difference lemma (Lemma B.1) and Lemma B.4

$$V^*(d_0) - \widetilde{V}^*(d_0) \le V^*(d_0) - \widetilde{V}^{\pi^*}(d_0)$$

$$= (1 - \lambda)\gamma \mathbb{E}_{\rho^{\pi^*}(d_0)} \left[ \sum_{t=1}^\infty (\lambda\gamma)^{t-1} (V^*(s_t) - h(s_t)) \right] = (1 - \lambda)\gamma \mathcal{C}(\pi, V^* - f, \lambda\gamma)$$

For the second term, we can rewrite it as

$$\frac{\gamma(1 - \lambda)}{1 - \gamma} \mathbb{E}_{s,a \sim d^\pi} \mathbb{E}_{s'|s,a} \left[ h(s') - \widetilde{V}^*(s') \right] = \gamma(1 - \lambda) \mathbb{E}_{\rho^\pi(d_0)} \left[ \sum_{t=1}^\infty \gamma^{t-1} (h(s_t) - \widetilde{V}^*(s_t)) \right]$$

$$= (1 - \lambda)\gamma \mathcal{C}(\pi^*, f - \widetilde{V}^*, \gamma)$$

## A.6 Proof of Proposition 4.4

**Proposition 4.4.** *If $h$ is improvable with respect to $\mathcal{M}$, then $\widetilde{V}^*(s) \ge h(s)$, for all $\lambda \in [0, 1]$.*

Let $d_t^\pi(s; s_0)$ denote the state distribution at the $t$th step after running $\pi$ starting from $s_0 \in \mathcal{S}$ in $\mathcal{M}$ (i.e. $d_0^\pi(s; s_0) = \mathbb{1}\{s = s_0\}$). Define the mixture

$$\widetilde{d}_{s_0}^\pi(s) := (1 - \widetilde{\gamma}) \sum_{t=0}^\infty \widetilde{\gamma}^t d_t^\pi(s; s_0) \tag{5}$$

where we recall the new discount $\widetilde{\gamma} = \gamma\lambda$ By performance difference lemma (Lemma B.1), we can write for any policy $\pi$ and any $s_0 \in$

$$\widetilde{V}^\pi(s_0) - h(s_0) = \frac{1}{1 - \lambda\gamma} \mathbb{E}_{\widetilde{d}_{s_0}^\pi} \left[ (\widetilde{\mathcal{B}}h)(s, a) - h(s) \right]$$

Notice that

$$(\widetilde{\mathcal{B}}h)(s, a) = \widetilde{r}(s, a) + \widetilde{\gamma} \mathbb{E}_{s'|s,a}[h(s')]$$
$$= r(s, a) + (1 - \lambda)\gamma \mathbb{E}_{s'|s,a}[h(s')] + \lambda\gamma \mathbb{E}_{s'|s,a}[h(s')]$$
$$= r(s, a) + \gamma \mathbb{E}_{s'|s,a}[h(s')] = (\mathcal{B}h)(s, a)$$

Let $\pi$ denote the greedy policy of $\arg\max_a (\mathcal{B}h)(s, a)$. Then we have, by the improvability assumption we have $(\mathcal{B}h)(s, \pi) - h(s) \ge 0$ and therefore,

$$\widetilde{V}^*(s_0) \ge \widetilde{V}^\pi(s_0) = h(s_0) + \frac{1}{1 - \lambda\gamma} \mathbb{E}_{\widetilde{d}_{s_0}^\pi} \left[ (\widetilde{\mathcal{B}}h)(s, a) - h(s) \right]$$

$$= h(s_0) + \frac{1}{1 - \lambda\gamma} \mathbb{E}_{\widetilde{d}_{s_0}^\pi} \left[ (\mathcal{B}h)(s, a) - h(s) \right]$$

$$\ge h(s_0)$$

Since $s_0$ is arbitrary above, we have the desired statement.

## A.7 Proof of Proposition 4.5

**Proposition 4.5.** *Suppose $h(s) = Q(s, \pi')$ for some policy $\pi'$ and function $Q : \mathcal{S} \times \mathcal{A} \to \mathbb{R}$ such that $Q(s, a) \le (\mathcal{B}h)(s, a)$, $\forall s \in \mathcal{S}, a \in \mathcal{A}$. Then $h$ is improvable and $f(s) \le V^{\pi'}(s)$ for all $s \in \mathcal{S}$.*

The proof is straightforward: We have $\max_a (\mathcal{B}h)(s,a) \geq (\mathcal{B}h)(s,\pi) \geq Q(s,\pi) = h(s)$, which is the definition of $h$ being improvable. For the argument of uniform lower bound, we chain the assumption $Q(s,a) \leq (\mathcal{B}h)(s,a)$:

$$
\begin{aligned}
h(s) = Q(s,\pi') &= r(s,\pi') + \gamma \mathbb{E}_{s'|s,\pi'}[h(s')] \\
&\leq r(s,\pi') + \gamma \left( r(s',\pi'), +\gamma \mathbb{E}_{s''|s',\pi'}[h(s'')] \right) \\
&\leq V^{\pi'}(s)
\end{aligned}
$$

# B Technical Lemmas

## B.1 Lemmas of Performance Difference

Here we prove a general performance difference for the $\lambda$-weighting used in the reshaped MDPs.

**Lemma A.1** (General Performance Difference Lemma). *Consider the reshaped MDP $\widetilde{\mathcal{M}}$ defined by some $f : \mathcal{S} \to \mathbb{R}$ and $\lambda \in [0,1]$. For any policy $\pi$, any state distribution $d_0$ and any $V : \mathcal{S} \to \mathbb{R}$, it holds that*

$$
\begin{aligned}
V(d_0) - V^\pi(d_0) &= \frac{\gamma(1-\lambda)}{1-\gamma} \mathbb{E}_{s,a\sim d^\pi} \mathbb{E}_{s'|s,a}\left[ h(s') - V(s') \right] \\
&\quad + \lambda \left( V(d_0) - \widetilde{V}^\pi(d_0) \right) + \frac{1-\lambda}{1-\gamma} \left( V(d^\pi) - \widetilde{V}^\pi(d^\pi) \right)
\end{aligned}
$$

Our new lemma includes the two below performance difference lemmas in the literature as special cases. Lemma B.2 can be obtained by setting $V = f$; Lemma B.1 can be obtained by further setting $\lambda = 0$ (that is, Lemma B.1 is a special case of Lemma B.2 with $\lambda = 0$; and Lemma A.1 generalizes both). The proofs of these existing performance difference lemmas do not depend on the new generalization in Lemma A.1, please refer to [17, 55] for details.

**Lemma B.1** (Performance Difference Lemma [17, 55] ). *For any policy $\pi$, any state distribution $d_0$ and any $V : S \to \mathbb{R}$, it holds that*

$$
V(d_0) - V^\pi(d_0) = \frac{1}{1-\gamma} \mathbb{E}_{d^\pi}\left[ V(s) - (\mathcal{B}V)(s,a) \right]
$$

**Lemma B.2** ($\lambda$-weighted Performance Difference Lemma [17]). *For any policy $\pi$, $\lambda \in [0,1]$, and $f : \mathcal{S} \to \mathbb{R}$, it holds that*

$$
f(d_0) - V^\pi(d_0) = \lambda \left( f(d_0) - \widetilde{V}^\pi(d_0) \right) + \frac{1-\lambda}{1-\gamma} \left( f(d^\pi) - \widetilde{V}^\pi(d^\pi) \right)
$$

### B.1.1 Proof of Lemma A.1

First, we use the standard performance difference lemma (Lemma B.1) in the original MDP and Lemma B.3 for the first and the last steps below,

$$
\begin{aligned}
V(d_0) - V^\pi(d_0) &= \frac{1}{1-\gamma} \mathbb{E}_{d^\pi}\left[ V(s) - (\mathcal{B}V)(s,a) \right] \\
&= \frac{1}{1-\gamma} \mathbb{E}_{d^\pi}\left[ (\widetilde{\mathcal{B}}V)(s,a) - (\mathcal{B}V)(s,a) \right] + \frac{1}{1-\gamma} \mathbb{E}_{d^\pi}\left[ V(s) - (\widetilde{\mathcal{B}}V)(s,a) \right] \\
&= \frac{\gamma(1-\lambda)}{1-\gamma} \mathbb{E}_{s,a\sim d^\pi} \mathbb{E}_{s'|s,a}\left[ h(s') - V(s') \right] + \frac{1}{1-\gamma} \mathbb{E}_{s,a\sim d^\pi}\left[ V(s) - (\widetilde{\mathcal{B}}V)(s,a) \right]
\end{aligned}
$$

Finally, substituting the equality in Lemma B.6 into the above equality concludes the proof.

## B.2 Properties of reshaped MDP

The first lemma is the difference of Bellman backups.

**Lemma B.3.** *For any $V : \mathcal{S} \to \mathbb{R}$,*

$$
(\mathcal{B}V)(s,a) - (\widetilde{\mathcal{B}}V)(s,a) = (1-\lambda)\gamma \mathbb{E}_{s'|s,a}[V(s') - h(s')]
$$

*Proof.* The proof follows from the definition of the reshaped MDP:

$$
\begin{aligned}
&(\mathcal{B}V)(s,a) - (\widetilde{\mathcal{B}}V)(s,a) \\
&= r(s,a) + \gamma \mathbb{E}_{s'|s,a}[V(s')] - r(s,a) - (1-\lambda)\gamma\mathbb{E}_{s'|s,a}[h(s')] - \gamma\lambda\mathbb{E}_{s'|s,a}[V(s')] \\
&= (1-\lambda)\gamma\mathbb{E}_{s'|s,a}[V(s') - h(s')]
\end{aligned}
$$

∎

This lemma characterizes, for a policy, the difference in returns.

**Lemma B.4.** *For any policy $\pi$ and $h : \mathcal{S} \to \mathbb{R}$,*

$$
V^\pi(s) - \widetilde{V}^\pi(s) = (1-\lambda)\gamma\mathbb{E}_{\rho^\pi(s)}\left[\sum_{t=1}^\infty (\lambda\gamma)^{t-1}(V^\pi(s_t) - h(s_t))\right]
$$

*Proof.* The proof is based on performance difference lemma (Lemma B.1) applied in the reshaped MDP and Lemma B.3. Recall the definition $\widetilde{d}_{s_0}^\pi(s)$ in (5) and define $\widetilde{d}_{s_0}^\pi(s,a) = \widetilde{d}_{s_0}^\pi(s)\pi(a|s)$. For any $s_0 \in \mathcal{S}$,

$$
\begin{aligned}
V^\pi(s_0) - \widetilde{V}^\pi(s_0) &= \frac{1}{1-\gamma\lambda}\mathbb{E}_{s,a\sim\widetilde{d}_{s_0}^\pi}[V^\pi(s) - \widetilde{\mathcal{B}}V^\pi(s,a)] \\
&= \frac{1}{1-\gamma\lambda}\mathbb{E}_{s,a\sim\widetilde{d}_{s_0}^\pi}[(\mathcal{B}V^\pi)(s,a) - (\widetilde{\mathcal{B}}V^\pi)(s,a)] \\
&= \frac{(1-\lambda)\gamma}{1-\gamma\lambda}\mathbb{E}_{s,a\sim\widetilde{d}_{s_0}^\pi}\mathbb{E}_{s'|s,a}[V^\pi(s') - h(s')]
\end{aligned}
$$

Finally, substituting the definition of $\widetilde{d}_{s_0}^\pi$ finishes the proof. ∎

A consequent lemma shows that $h$ and $\widetilde{V}^\pi$ are close, when $h$ and $V^\pi$ are.

**Lemma B.5.** *For a policy $\pi$, suppose $-\epsilon_l \le h(s) - V^\pi(s) \le \epsilon_u$. It holds*

$$
-\epsilon_l - \frac{(1-\lambda)\gamma\epsilon_u}{1-\lambda\gamma} \le h(s) - \widetilde{V}^\pi(s) \le \epsilon_u + \frac{(1-\lambda)\gamma\epsilon_l}{1-\lambda\gamma}
$$

*Proof.* We prove the upper bound by Lemma B.4; the lower bound can be shown by symmetry.

$$
\begin{aligned}
h(s) - \widetilde{V}^\pi(s) &\le \epsilon_u + V^\pi(s) - \widetilde{V}^\pi(s) \\
&= \epsilon_u + (1-\lambda)\gamma\mathbb{E}_{\rho^\pi(s)}\left[\sum_{t=1}^\infty (\lambda\gamma)^{t-1}(V^\pi(s_t) - h(s_t))\right] \\
&\le \epsilon_u + \frac{(1-\lambda)\gamma\epsilon_l}{1-\lambda\gamma}
\end{aligned}
$$

∎

The next lemma relates online Bellman error to value gaps.

**Lemma B.6.** *For any $\pi$ and $V : \mathcal{S} \to \mathbb{R}$,*

$$
\frac{1}{1-\gamma}\left(\mathbb{E}_{d^\pi}\left[V(s) - (\widetilde{\mathcal{B}}V)(s,a)\right]\right) = \lambda\left(V(d_0) - \widetilde{V}^\pi(d_0)\right) + \frac{1-\lambda}{1-\gamma}\left(V(d^\pi) - \widetilde{V}^\pi(d^\pi)\right)
$$

*Proof.* We use Lemma B.3 in the third step below.

$$\mathbb{E}_{d^\pi}\left[V(s) - (\widetilde{\mathcal{B}}V)(s,a)\right]$$

$$= \mathbb{E}_{d^\pi}\left[V(s) - (\widetilde{\mathcal{B}}\widetilde{V}^\pi)(s,a)\right] + \mathbb{E}_{d^\pi}\left[\widetilde{\mathcal{B}}\widetilde{V}^\pi(s,a) - (\widetilde{\mathcal{B}}V)(s,a)\right]$$

$$= \mathbb{E}_{d^\pi}\left[V(s) - \widetilde{V}^\pi(s)\right] + \mathbb{E}_{d^\pi}\left[(\widetilde{\mathcal{B}}\widetilde{V}^\pi)(s,a) - (\widetilde{\mathcal{B}}V)(s)\right]$$

$$= \mathbb{E}_{d^\pi}\left[V(s) - \widetilde{V}^\pi(s)\right] - \lambda\gamma\mathbb{E}_{s,a\sim d^\pi}\mathbb{E}_{s'|s,a}\left[\widetilde{V}^\pi(s') - V(s')\right]$$

$$= (1-\gamma)\mathbb{E}_{\rho^\pi(d_0)}\left[\sum_{t=0}^{\infty}\gamma^t(V(s_t) - \widetilde{V}^\pi(s_t)) - \lambda\gamma^{t+1}(\widetilde{V}^\pi(s_{t+1}) - V(s_{t+1}))\right]$$

$$= (1-\gamma)\lambda(V(d_0) - \widetilde{V}^\pi(d_0)) + (1-\gamma)(1-\lambda)\mathbb{E}_{\rho_\pi(d_0)}\left[\sum_{t=0}^{\infty}\gamma^t(V(s_t) - \widetilde{V}^\pi(s_t))\right]$$

∎

# C  Experiments

## C.1  Details of the MuJoCo Experiments

We consider four dense reward MuJoCo environments (Hopper-v2, HalfCheetah-v2, Humanoid-v2, and Swimmer-v2) and a sparse reward version of Reacher-v2.

In the sparse reward Reacher-v2, the agent receives a reward of 0 at the goal state (defined as $\|g(s) - e(s)\| \leq 0.01$ and $-1$ elsewhere, where $g(s)$ and $e(s)$ denote the goal state and the robot's end-effector positions, respectively. We designed a heuristic $h(s) = r(s,a) - 100\|e(s) - g(s)\|$, as this is a goal reaching task. Here the policy is randomly initialized, as no prior batch data is available before interactions.

In the dense reward experiments, we suppose that a batch of data collected by multiple behavioral policies are available before learning, and a heuristic is constructed by an offline policy evaluation algorithm from the batch data; in the experiments, we generated these behavioral policies by running SAC from scratch and saved the intermediate policies generated in training. We designed this heuristic generation experiment to simulate the typical scenario where offline data collected by multiple policies of various qualities is available before learning. In this case, a common method for inferring what values a good policy could get is to inspect the realized accumulated rewards in the dataset. For simplicity, we use basic Monte Carlo regression to construct heuristics, where a least squares regression problem was used to fit a fully connected neural network to predict the empirical returns on the trajectories in the sampled batch of data.

Specifically, for each dense reward Mujoco experiment, we ran SAC for 200 iterations and logged the intermediate policies for every 4 iterations, resulting in a total of 50 behavior policies. In each random seed of the experiment, we performed the following: We used each behavior policy to collect trajectories of at most 10,000 transition tuples, which gave about 500,000 offline data points over these 50 policies. These data were used to construct the Monte-Carlo regression data, which was done by computing the accumulated discounted rewards along sampled trajectories. Then we generated the heuristic used in the experiment by fitting a fully connected NN with (256,256)-hidden layers using default ADAM with step size 0.001 and minibatch size 128 for 30 epochs over this randomly generated dataset of 50 behavior policies.

For the dense reward Mujoco experiments, we also use behavior cloning (BC) with $\ell_2$ loss to warm start RL agents based on the same batch dataset of 500,000 offline data points. The base RL algorithm here is SAC, which is based on the standard implementation of Garage (MIT License) [37]. The policy and the value networks are fully connected neural networks, independent of each other. The policy is Tanh-Gaussian and the value network has a linear head.

**Algorithms.**  We compare the performance of different algorithms below. *1)* BC *2)* SAC *3)* SAC with BC warm start (SAC w/ BC) *4)* HuRL with a zero heuristic and BC warm start (HuRL-zero)

*5)* HuRL with the Monte-Carlo heuristic and BC warm start (HuRL-MC). For the HuRL algorithms, the mixing coefficient $\lambda_n$ is scheduled as

$$\lambda_n = \lambda_0 + (1 - \lambda_0) \tanh\left( \frac{n-1}{\alpha N - 1} \times \arctan(0.99) \right) / 0.99$$

$$=: \lambda_0 + (1 - \lambda_0) c_\omega \tanh(\omega(n-1))$$

for $n = 1, \ldots, N$, where $\lambda_0 \in [0, 1]$ is the initial $\lambda$ and $\alpha > 0$ controls the increasing rate. This schedule ensures that $\lambda_N = 1$ when $\alpha = 1$. Increasing $\alpha$ from 1 makes $\lambda_n$ converge to 1 slower.

We chose these algorithms to illustrate the effect of each additional warm-start component (BC and heuristics) added on top of the base algorithm SAC. HuRL-zero is SAC w/ BC but with an extra $\lambda$ schedule described above that further lowers the discount, whereas SAC and SAC w/ BC keep a constant discount factor.

**Evaluation and Hyperparameters.**   In each iteration, the RL agent has a fixed sample budget for environment interactions, and its performance is measured in terms of the undiscounted accumulated rewards (estimated by 10 rollouts) of the deterministic mean policy extracted from SAC. The hyperparameters used in the algorithms above were selected as follows. The selection was done by uniformly random grid search[7] over the range of hyperparameters in Table 1 to maximize the AUC of the training curve.

| | |
|---|---|
| Polcy step size | [0.00025, 0.0005, 0.001, 0.002] |
| Value step size | [0.00025, 0.0005, 0.001, 0.002] |
| Target step size | [0.005, 0.01, 0.02, 0.04] |
| $\gamma$ | [0.9, 0.99, 0.999] |
| $\lambda_0$ | [0.90, 0.95, 0.98, 0.99] |
| $\alpha$ | [$10^{-5}$, 1.0, $10^5$] |

Table 1: HuRL's hyperparameter value grid for the MuJoCo experiments.

First, the learning rates (policy step size, value step size, target step size) and the discount factor of the base RL algorithm, SAC, were tuned for each environment to maximize the performance. This tuned discount factor is used as the de facto discount factor $\gamma$ of the original MDP $\mathcal{M}$. Fixing the hyperparameters above, $\lambda_0$ and $\alpha$ for the $\lambda$ schedule of HuRL were tuned for each environment and each heuristic. The tuned hyperparameters and the environment specifications are given in Tables 2 and 3 below. (The other hyperparameters, in addition to the ones tuned above, were selected manually and fixed throughout all the experiments).

Finally, after all these hyperparameters were decided, we conducted additional testing runs with 30 different random seeds and report their statistics here. The randomness include the data collection process of the behavioral policies, training the heuristics from batch data, BC, and online RL, but the behavioral policies are fixed.

While this procedure takes more compute (the computation resources are reported below; tuning the base SAC takes the most compute), it produces more reliable results without (luckily or unluckily) using some hand-specified hyperparameters or a particular way of aggregating scores when tuning hyperparameters across environments. Empirically, we also found using constant $\lambda$ around $0.95 \sim 0.98$ leads to good performance, though it may not be the best environment-specific choice.

**Resources.**   Each run of the experiment was done using an Azure Standard_H8 machine (8 Intel Xeon E5 2667 CPUs; memory 56 GB; base frequency 3.2 GHz; all cores peak frequency 3.3 GHz; single core peak frequency 3.6 GHz). The Hopper-v2, HalfCheetah-v2, Swimmer-v2 experiments took about an hour per run. The Humanoid-v2 experiments took about 4 hours. No GPU was used.

**Extra Experiments with VAE-based Heuristics.**   We conduct additional experiments of HuRL using a VAE-filtered pessimistic heuristic. This heuristic is essentially the same as the Monte-Carlo

---

[7]We ran 300 and 120 randomly chosen configurations from Table 1 with different random seeds to tune the base algorithm and the $\lambda$-scheduler, respectively. Then the best configuration was used in the following experiments.

| Environment | Sparse-Reacher-v2 |
|---|---|
| Obs. Dim | 11 |
| Action Dim | 2 |
| Evaluation horizon | 500 |
| $\gamma$ | 0.9 |
| Batch Size | 10000 |
| Policy NN Architecture | (64,64) |
| Value NN Architecture | (256,256) |
| Polcy step size | 0.00025 |
| Value step size | 0.00025 |
| Target step size | 0.02 |
| Minibatch Size | 128 |
| Num. of Grad. Step per Iter. | 1024 |
| HuRL $\lambda_0$ | 0.5 |
| HuRL-MC $\alpha$ | $10^5$ |

Table 2: Sparse reward MuJoCo experiment configuration details. All the values other than $\lambda$-scheduler's (i.e. those used in SAC) are shared across different algorithms in the comparison. All the neural networks here fully connected and have `tanh` activation; the numbers of hidden nodes are documented above. Note that when $\alpha = 10^5$, effectively $\lambda_n = \lambda_0$ in the training iterations; when $\alpha = 10^{-5}$, $\lambda_n \approx 1$ throughout.

| Environment | Hopper-v2 | HalfCheetah-v2 | Swimmer-v2 | Humanoid-v2 |
|---|---|---|---|---|
| Obs. Dim | 11 | 17 | 8 | 376 |
| Action Dim | 3 | 6 | 2 | 17 |
| Evaluation horizon | 1000 | 1000 | 1000 | 1000 |
| $\gamma$ | 0.999 | 0.99 | 0.999 | 0.99 |
| Batch Size | 4000 | 4000 | 4000 | 10000 |
| Policy NN Architecture | (64,64) | (64,64) | (64,64) | (256,256) |
| Value NN Architecture | (256,256) | (256,256) | (256,256) | (256,256) |
| Polcy step size | 0.00025 | 0.00025 | 0.0005 | 0.002 |
| Value step size | 0.0005 | 0.0005 | 0.0005 | 0.00025 |
| Target step size | 0.02 | 0.04 | 0.0100 | 0.02 |
| Num. of Behavioral Policies | 50 | 50 | 50 | 50 |
| Minibatch Size | 128 | 128 | 128 | 128 |
| Num. of Grad. Step per Iter. | 1024 | 1024 | 1024 | 1024 |
| HuRL-MC $\lambda_0$ | 0.95 | 0.99 | 0.95 | 0.9 |
| HuRL-MC $\alpha$ | $10^5$ | $10^5$ | 1.0 | 1.0 |
| HuRL-zero $\lambda_0$ | 0.98 | 0.99 | 0.99 | 0.95 |
| HuRL-zero $\alpha$ | $10^{-5}$ | $10^5$ | 1.0 | $10^{-5}$ |

Table 3: Dense reward MuJoCo experiment configuration details. All the values other than $\lambda$-scheduler's (i.e. those used in SAC) are shared across different algorithms in the comparison. All the neural networks here fully connected and have `tanh` activation; the numbers of hidden nodes are documented above. Note that when $\alpha = 10^5$, effectively $\lambda_n = \lambda_0$ in the training iterations; when $\alpha = 10^{-5}$, $\lambda_n \approx 1$ throughout.

regression-based heuristic we discussed, except that an extra VAE (variational auto-encoder) is used to classify states into known and unknown states in view of the batch dataset, and then the predicted values of unknown states are set to be the lowest empirical return seen in the dataset. In implementation, this is done by training a state VAE (with a latent dimension of 32) to model the states in the batch data, and then a new state classified as unknown if its VAE loss is higher than 99-th percentile of the VAE losses seen on the batch data. The implementation and hyperparameters are based on the code from Liu et al. [31]. We note, however, that this basic VAE-based heuristic does not satisfy the assumption of Proposition 4.5.

These results are shown in Fig. 3, where HuRL-VAEMC denotes HuRL using this VAE-based heuristic. Overall, we see that such a basic pessimistic estimate does not improve the performance from the pure Monte-Carlo version (HuRL-MC); while it does improve the results slightly in

HalfCheetah-v2, it gets worse results in Humanoid-v2 and Swimmer-v2 compared with HuRL-MC. Nonetheless, HuRL-VAEMC is still better than the base SAC.

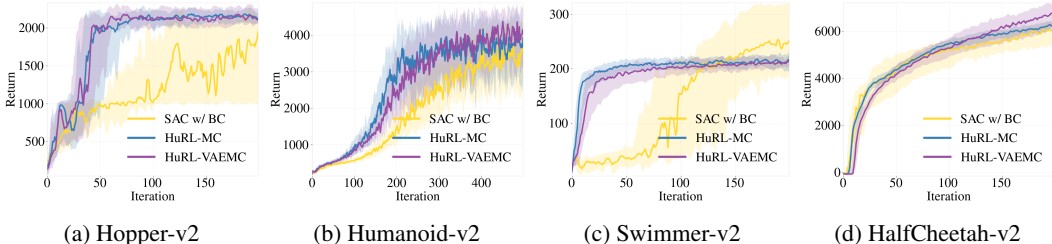

(a) Hopper-v2          (b) Humanoid-v2          (c) Swimmer-v2          (d) HalfCheetah-v2

Figure 3: Extra experimental results of different MuJoCo environments. The plots show the 25th, 50th, 75th percentiles of each algorithm's performance over 30 random seeds.

## C.2 Procgen Experiments

In addition to MuJoCo environments, where the agent has direct access to the true low-dimensional system state, we conducted experiments on the Procgen benchmark suite [33, 56]. The Procgen suite consists of 16 procedurally generated Atari-like game environments, whose main conceptual differences from MuJoCo environments are partial observability and much higher dimensionality of agents' observations (RGB images). The 16 games are very distinct structurally, but each game has an unlimited number of levels[8] that share common characteristics. All levels of a given game are situated in the same underlying state space and have the same transition function but differ in terms of the regions of the state space reachable within each level and in their observation spaces. We focus on the *sample efficiency* Procgen mode [33]: in each RL episode the agent faces a new game level, and is expected to eventually learn a single policy that performs well across all levels of the given game.

Besides the differences in environment characteristics between MuJoCo and Procgen, the Procgen experiments are also dissimilar in their design:

- In contrast to the MuJoCo experiments, where we assumed to be given a batch of data from which we constructed a heuristic and a warm-start policy, in the Procgen experiments we simulate a scenario where we are given *only* the heuristic function itself. Thus, we don't warm-start the base algorithm with a BC policy when running HuRL.

- In the Procgen experiments, we share a single set of all hyperparameters' values – those of the base algorithm, those of HuRL's $\lambda$-scheduling, and those used for generating heuristics – across all 16 games. This is meant to simulate a scenario where HuRL is applied across a diverse set of problems using good but problem-independent hyperparameters.

**Algorithms.** We used PPO [36] implemented in RLlib (Apache License 2.0) [57] as the base algorithm. We generated a heuristic for each game as follows:

- We ran PPO for $8M$ environment interaction steps and saved the policy after every $500K$ steps, for a total of 16 checkpoint policies.

- We ran the policies in a random order by executing 12000 environment interaction steps using each policy. For each rollout trajectory, we computed the discounted return for each observation in that trajectory, forming $\langle observation, return \rangle$ training pairs.

- We used this data to learn a heuristic via regression. We mixed the data, divided it into batches of 5000 training pairs and took a gradient step w.r.t. MSE computed over each batch. The learning rate was $10^{-4}$.

Our main algorithm, a HuRL flavor denoted as PPO-HuRL, is identical to the base PPO but uses the Monte-Carlo heuristic computed as above.

---

[8]In Procgen, levels aren't ordered by difficulty. They are merely game variations.

**Hyperparameters and evaluation**   The base PPO's hyperparameters in RLlib were chosen to match PPO's performance reported in the original Procgen paper [56] for the "easy" mode as closely as possible across all 16 games (Cobbe et al. [56] used a different PPO implementation with a different set of hyperparameters). As in that work, our agent used the IMPALA-CNN$\times$4 network architecture [56, 58] without the LSTM. The heuristics employed the same architecture as well. We used a single set of hyperparameter values, listed in Table 4, for all Procgen games, both for policy learning and for generating the checkpoints for computing the heuristics.

| | |
|---|---|
| Impala layer sizes | 16, 32, 32 |
| Rollout fragment length | 256 |
| Number of workers | 0 *(in RLlib, this means 1 rollout worker)* |
| Number of environments per worker | 64 |
| Number of CPUs per worker | 5 |
| Number of GPUs per worker | 0 |
| Number of training GPUs | 1 |
| $\gamma$ | 0.99 |
| SGD minibatch size | 2048 |
| Train batch size | 4000 |
| Number of SGD iterations | 3 |
| SGD learning rate | 0.0005 |
| Framestacking | off |
| Batch mode | truncate_episodes |
| Value function clip parameter | 10.0 |
| Value function loss coefficient | 0.5 |
| Value function share layers | true |
| KL coefficient | 0.2 |
| KL target | 0.01 |
| Entropy coefficient | 0.01 |
| Clip parameter | 0.1 |
| Gradient clip | null |
| Soft horizon | False |
| No done at end: | False |
| Normalize actions | False |
| Simple optimizer | False |
| Clip rewards | False |
| GAE $\lambda$ | 0.95 |
| PPO-HuRL $\lambda_0$ | 0.99 |
| PPO-HuRL $\alpha$ | 0.5 |

Table 4: Procgen experiment configuration details: RLlib PPO's and HuRL's hyperparameter values. All the values were shared across all 16 Procgen games.

| | |
|---|---|
| $\lambda_0$ | [0.95, 0.97, 0.985, 0.98, 0.99] |
| $\alpha$ | [0.5, 0.75, 1.0, 3.0, 5.0] |

Table 5: HuRL's hyperparameter value grid for the Procgen experiments.

In order to choose values for PPO-HuRL's hyperparameters $\alpha$ and $\lambda_0$, we fixed all of PPO's hyperparameters, took the pre-computed heuristic for each game, and did a grid search over $\alpha$ and $\lambda_0$'s values listed in Table 5 to maximize the normalized average AUC across all games. To evaluate each hyperparameter value combination, we used 4 runs per game, each run using a random seed and lasting 8M environment interaction steps. The resulting values are listed in Table 4. Like PPO's hyperparameters, they were kept fixed for all Procgen environments.

To obtain experimental results, we ran PPO and PPO-HuRL with the aforementioned hyperparameters on each of 16 games 20 times, each run using a random seed and lasting 8M steps as in Mohanty et al. [33]. We report the 25th, 50th, and 75th-percentile training curves. Each of the reported training curves was computed by smoothing policy performance in terms of unnormalized game scores over the preceding 100 episodes.

**Resources.** Each policy learning run used a single Azure ND6s machine (6 Intel Xeon E5-2690 v4 CPUs with 112 GB memory and base core frequency of 2.6 GHz; 1 P40 GPU with 24 GB memory). A single PPO run took approximately 1.5 hours on average. A single PPO-HuRL run took approximately 1.75 hours.

**Results.** The results are shown in Fig. 4. They indicate that, HuRL helps despite the highly challenging setup of this experiment: a) environments with a high-dimensional observation space; a) the chosen hyperparameter values being likely suboptimal for individual environments; c) the heuristics naively generated using Monte-Carlo samples from a mixture of policies of wildly varying quality; and d) the lack of policy warm-starting. We hypothesize that PPO-HuRL's performance can be improved further with environment-specific hyperparameter tuning and a scheme for heuristic-quality-dependent adjustment of HuRL's $\lambda$-schedules on the fly.

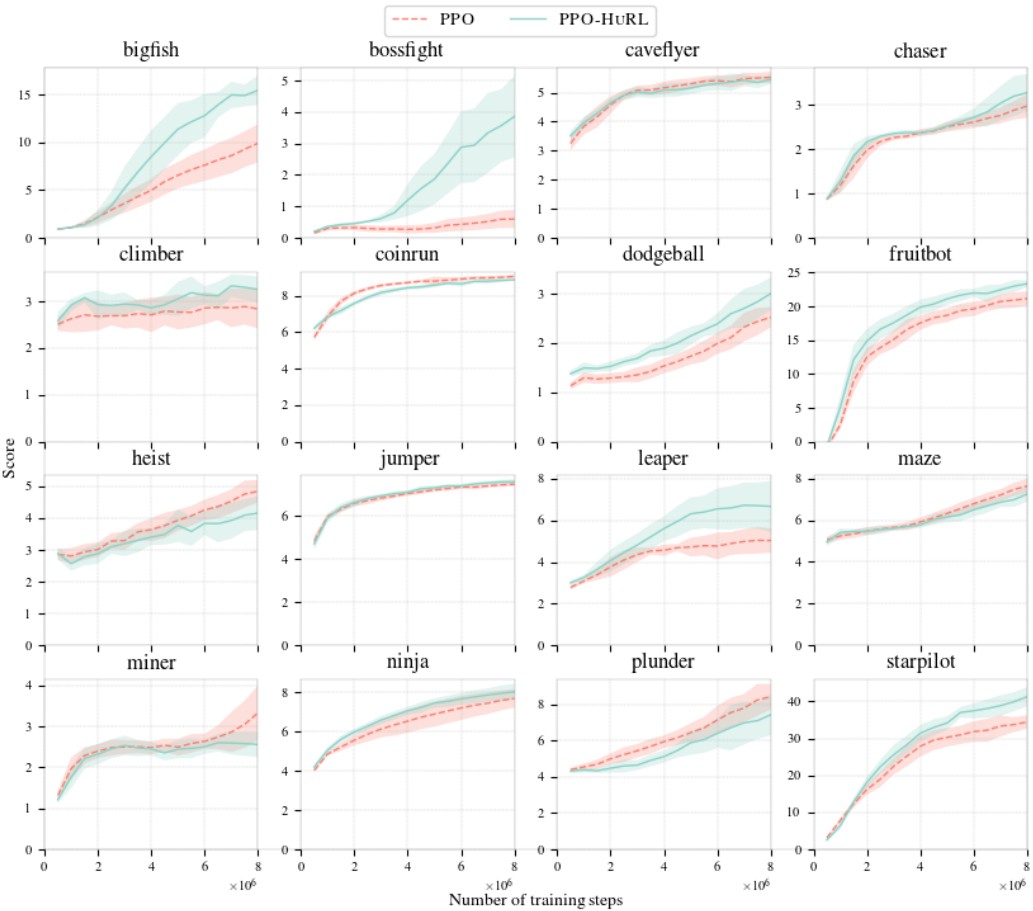

Figure 4: PPO-HuRL's results on Procgen games. PPO-HuRL yields gains on half of the games and performs at par with PPO on most of the rest. Thus, on balance, PPO-HuRL helps despite the highly challenging setup of this experiment, but tuning HuRL's $\lambda$-schedule on the fly depending on the quality of the heuristic can potentially make HuRL's performance more robust in settings like this.