# OpenReview forum: "Heuristic-Guided Reinforcement Learning"
_NeurIPS.cc/2021/Conference — NeurIPS 2021 Poster_

### Official Review · Reviewer_vrFt · 2021-06-30

**Rating:** 6
**Confidence:** 3

**Summary:**

This work proposes a framework for incorporating heuristics into reinforcement learning. The main idea is to construct a new MDP where the new reward function is offset by a scaled heuristic and the new temporal discount factor is scaled down. The scaling of the heuristic and temporal discount factor are controlled by a mixing coefficient $\lambda \in [0, 1]$. The proposal is to run an RL algorithm in this new MDP with the mixing coefficient adjusted over time (HuRL). The main theoretical result relates performance in the new MDP to performance in the original MDP (Theorem 4.1). Empirical results in MuJoCo and Procgen with PPO and SAC show that HuRL improves on several baselines.

**Limitations And Societal Impact:**

See “concerns and critiques” above.

**Main Review:**


Overall positive comments:

* To my knowledge, the MDP reduction proposed in this work has not been previously considered in the literature.
* The inclusion of code with the submission is appreciated.
* Reporting results with 30 random seeds is commendable!
* The paper is relatively well organized.
* The related work section covers the main areas that I would consider relevant.
* I did not find any major errors in the theoretical analysis.
* Proposition 2.4 is quite interesting -- I wish there were more discussion and focus on that result.


Critiques and concerns:


* My main concern is significance with respect to potential-based reward shaping (PBRS). As a practitioner, I am not sure why I would opt for HuRL instead of PBRS.

The main benefit of PBRS is that it is optimality-preserving, so there is no need to worry about introducing Bias, like there is with HuRL. Another benefit of PBRS is that it does not require tuning any of the additional hyperparameters that are introduced in HuRL.

The paper mentions two possible reasons to prefer HuRL:

(1) “PBRS does not fully leverage information provided by the heuristic because it still requires the learning agent to solve a long horizon problem, which is the main source of difficulty in RL. Conceptually, when the heuristic is the optimal value function, the agent should be able to find the optimal policy by acting myopically, as $V^*$ already contains all necessary long-term information for good decision making.” [L103-108]


(2) “We choose (2) over the PBRS version, because the latter does not preserve the properties in Proposition 4.2.” [Footnote 3]

The second reason is interesting, but requires more discussion and support (e.g. proof that PBRS does not have the properties in proposition 4.2). The first reason is presented as the main motivation for not using PBRS, but I do not find it convincing. For example, in the original PBRS paper, at the end of their Section 3, they discuss how reward shaping using the optimal value function $V^*$ would work particularly well for PBRS, saying that all that would be left to do is determine Q values from V, which could be done with only a one-step lookahead (myopically). If I am misunderstanding the claims of L103-108, please let me know in the rebuttal, ideally with a concrete example.


Finally, PBRS should be included as a baseline in the experiments. This is an important empirical omission.
   * Relatedly, the claims that heuristics have not been considered very much in RL (e.g. L99-100) is not very convincing if we understand heuristics to be essentially the same as reward shaping, on which there has been a lot of work.
   * The environments and methods considered in the experiments are standard and sensible, but I would have liked to see HuRL applied in a setting where RL without heuristics is intractable and some good heuristics are available. One interesting possibility would be to borrow from the classical planning literature and use domain-independent PDDL heuristics (e.g., with PDDLGym).
   * Also, given the origin of the heuristics in the experiments, and the discussion of offline RL throughout the paper, it would be good to see offline RL baselines.
   * The fact that the HuRL-specific hyperparameters need to be tuned per-environment is limiting.
   * It looks like HuRL may be prematurely plateauing in Figure 2(c) [swimmer], can you comment on that?


Minor:
   * L22: manifests -> manifest
   * L74: wlog, reward function bounded between [0, 1]. This is not really wlog -- boundedness of the reward is an additional assumption.
   * L219: Regret(f,0,\pi) should be Regret(f,1,\pi)

**Time Spent Reviewing:**

5

---

> ### Author Response · Authors · 2021-08-11
> **Response to reviewer #vrFt**
>
> ### **Why prefer HuRL over PBRS**
>
> Please see the **PBRS vs. HuRL for policy *learning*.** section of the **Common response to all reviewers** comment. The key insight here is that, although optimality preservation, the ability to find $V^*$ from a (near-)optimal heuristic *if* the agent acted greedily, and PBRS’s other nice properties are useful for planning, they are not so much for reinforcement *learning* algorithms. They aren’t very helpful in RL because typical RL agents don’t actually do myopic lookahead. They initially don’t know the dynamics and hence successor states, so they try to estimate value functions by exploration, sampling and learning even if you give them a great estimate of long-term return (a heuristic) to begin with. In *learning*, HuRL’s interplay with bias matters, which is why HuRL is more useful in RL.
>
> This doesn’t mean that the PBRS paper is wrong -- it just gives intuitions that aren’t very applicable in RL (as opposed to planning).
>
> ### **Proposition 4.2**
>
> To see the benefit of Eq.(2) over the PBRS formulation, please note that in the PBRS version, $\bar{r}(s,a) = r(s,a) - f(s) + \gamma E_{s’|s,a}[f(s’)]$, the term $- f(s)$ can largely change the reward structure. For example, when $f(s)$ is in $ [0, 1/(1-\gamma)]$, it is possible that $\bar{r}(s,a)$ can take negative value as large as $-1/(1-\gamma)$ even when $r(s,a)$ in $[0,1]$, because $f(s)$ generally would not satisfy the Bellman equation of some policy. We can easily see that $\bar{r}(s,a)$ would destroy the linear MDP structure, when $f(s)$ does not belong to the linear class; on the other hand, Eq. (2) preserves linearity regardless of $f(s)$ for linear MDPs.
>
> Lastly, we can see why Eq. (2) is preferred over the PBRS version from a structured regression perspective. Suppose we would like to learn a Q function for $\bar{r}$ above and a policy $\pi$. That is, we wish to find Q such that $Q(s,a) = \bar{r}(s,a) + \gamma\lambda \mathbf{E}\_{s’|s,a}[Q(s’,\pi)]$ for some policy $\pi$. Since we know $\bar{r}(s,a) = r(s,a) - f(s) + \gamma \mathbf{E}\_{s’|s,a}[f(s’)]$  and $f(s)$ is known, we might consider leveraging this prior knowledge and parameterize $Q(s,a)$ as $Q(s,a) = F(s,a) - f(s)$, where $F(s,a)$ is the function to be learned. Substituting such structured $Q$ class into the above Bellman equation, we see that
>
> $Q(s,a) = F(s,a) - f(s) = \bar{r}(s,a) + \gamma\lambda \mathbf{E}\_{s’|s,a}[ Q(s’,\pi)] =
> r(s,a) - f(s) + \gamma \mathbf{E}\_{s’|s,a}[f(s’)]  + \gamma\lambda \mathbf{E}\_{s’|s,a}[ F(s’,\pi) - f(s’)]$
>
> which is
>
> $F(s,a) = r(s,a)  + \gamma (1-\lambda) \mathbf{E}\_{s’|s,a}[f(s')] + \gamma \lambda \mathbf{E}\_{s’|s,a}[F(s’)]
>           = \tilde{r}(s,a) + \gamma\lambda \mathbf{E}\_{s’|s,a}[F(s’)] $.
>
> In other words, we can view the HuRL with Eq. (2) as using a structured Q-class for the PBRS-version of HuRL. Using a structured class in learning usually reduces the unknown that needs to be learned and would lead to faster learning in practice.
>
> We will provide more discussions on this in the revision.
>
>
> ### **Claims that heuristics have not been considered very much in RL**
>
> We will rephrase these claims to emphasize that PBRS can be considered a flavor of learning with heuristics too and that there has been a lot of work on it.
>
> ### **Empirical comparison against PBRS.**
>
> We now realize that this was a significant omission and have performed such a comparison. Please see the description and results in the  **Empirical comparison of HuRL and PBRS.** section of the **Common response to all reviewers** comment. We will update the paper to include them.
>
>
> ### **Experiments with a Sparse Reward Problem**
>
> We conducted additional experiments in a sparse reward problem. Please see **HuRL with Heuristics based on Domain Knowledge** section of the **Common response to all reviewers** comment. It is in the type of setting you mentioned -- intractable with a heuristic, but a heuristic is easy to construct.
>
>
> ### **Offline RL baselines**
>
> Please note that HuRL is designed to work with *online* RL algorithms. The theory of HuRL does not make assumptions on the source of the heuristics -- this theory quantifies them only in terms of their approximation qualities. We do not assume the availability of offline data to the RL agent -- in the experiments, the offline data was used only to synthesize the heuristics, in order to simulate real-world applications where such data might be available, but we aren’t necessarily advocating the acquisition of heuristics in this way. In other words, offline RL is not an alternative to HuRL.
>
>
> ### **Hyperparameter selection**
>
> This is actually not necessary -- we did it only for a cleaner empirical comparison. Please see the  **HuRL and hyperparameter selection** section of the **Common response to all reviewers** comment.
>
>
> ### **Plateau in Figure 2(c)**
>
> We agree that plateauing is premature here. We suspect it’s due to the fact that SAC is not really an exploratory algorithm (like those based on bonuses) as well as the fact that we are representing policies and value functions with DNNs. As a result, even when $\lambda$ goes to 1 eventually, there is still a chance that SAC can get stuck in a local minimum.
>
>
> ### **Other remarks**
>
> Thank you for pointing out typos and ways to improve the writing. We will incorporate this into the revision.

---

> > ### Comment · Reviewer_vrFt · 2021-08-16
> > **Thanks for your responses**
> >
> > Thank you very much for your responses to my comments and questions, and thank you for the running the additional experiments. Regarding the new PBRS baseline, I am surprised that it performs so poorly compared to SAC alone. You mentioned reward re-scaling and side-information through $\lambda$ as two possible explanations for SAC+HuRL > SAC+PBRS. Reward re-scaling could explain SAC > SAC+PBRS too, but I would be surprised if it had such a dramatic effect.
> >
> > Is there a way to quantify the magnitude of reward re-scaling? Or to otherwise confirm your hypothesis that reward re-scaling explains the discrepancy in performance?
> >
> > Could you also confirm exactly the reward function transformation that you used to implement PBRS with SAC (either with math or a few lines of code)?
> >
> > I think my other questions and points were addressed well in the response. If we can confirm that the PBRS baseline is correct and fair then I will be willing to raise my score.

---

> > > ### Author Response · Authors · 2021-08-20
> > > **PBRS code and reward rescaling plots**
> > >
> > > Thanks for carefully reading our response!
> > >
> > > ### **Could you also confirm exactly the reward function transformation that you used to implement PBRS with SAC**
> > >
> > > Our implementation of PBRS is based on the code submitted with the paper, namely on the implementation of HuRL, which modifies the reward that SAC uses. The change for PBRS is only a few lines of code in the **reshape_rewards** method in *src/garage/shortrl/algos/sac.py* in the submitted code, which we include below:
> > >
> > > ```
> > > def reshape_rewards(self, rewards, next_obs, terminals, obs):
> > >       if self._reward_shaping_mode=='pbrs':
> > >       	hs = self._heuristic(obs)
> > >       	hs = hs[...,np.newaxis]
> > >       hs_next = self.heuristic(next_obs, terminals)
> > >       if self._reward_shaping_mode=='hurl':
> > >         assert rewards.shape == hs_next.shape == terminals.shape
> > >         return rewards + (self._discount0-self._discount)*hs_next
> > >       elif self._reward_shaping_mode=='pbrs':
> > >         assert rewards.shape == hs_next.shape == terminals.shape == hs.shape
> > >         return rewards + self._discount0*hs_next - hs
> > > ```
> > > Here self._discount0 is the original discount $\gamma$ in the paper, and self._discount is the guidance discount $\gamma\lambda$.
> > >
> > > Thus, the PBRS formula we have implemented is $\bar{r}(s,a) = r(s,a) + \gamma f(s') - f(s)$.
> > >
> > >
> > > ### **Is there a way to quantify the magnitude of reward re-scaling? Or to otherwise confirm your hypothesis that reward re-scaling explains the discrepancy in performance?**
> > >
> > > Yes, and to do this, we have gathered more data, presented in the figures here:
> > >
> > > https://figshare.com/s/7f8f7232274e0f9a4c43
> > >
> > > What the first 5 figures in the top row show is by how much SAC+PBRS and SAC+HuRL rescale rewards that are used for model updates as learning progresses in each of the 5 environments we’ve used so far (4 already in the paper + Sparse Reacher, presented earlier in the review responses). These figures are juxtaposed against the learning curves taken from the submitted paper and our earlier responses (the first 5 figures in the bottom row).
> > >
> > > Namely, for each of the 5 environments, we ran the same training as before (with 30 seeds), but this time we recorded, for each training iteration, **the ratio of absolute PBRS rewards to the absolute original rewards (those that SAC would use) averaged over the transitions sampled for updating the model in that iteration.** Specifically, we compute the average of $ |\bar{r}(s,a)| /(10^{-7}+ |r(s,a)|)$ for each iteration independently, where $\bar{r}$ denotes the reshaped reward. We recorded the same data for HuRL as well. The plots you see in the top row of figures show this ratio as a function of the number of training iterations. For Sparse Reacher, we show the plots both on the same scale and also plotted separately (the last figure in the top and in the bottom row) in order to show how “jittery” each plot is.
> > >
> > > The trend that immediately stands out in these plots is that **PBRS tends to cause erratic rescaling of rewards that SAC uses for updates from iteration to iteration and across different random seeds, as evidenced by the “jitter” in PBRS’s plots**. Our hypothesis is that this erratic rescaling is a source that destabilizes the learning of SAC+PBRS and makes its performance have high variance across seeds. HuRL, despite also inducing large *absolute* variations in rescaling throughout training, generally does this much more gradually.
> > >
> > > As a side note, in case you were wondering why the absolute rescaling of rewards is so high on Reacher, it is because it is a sparse-reward problem whose original rewards are zero most of the time. Therefore, the computed ratio is $|\bar{r}(s,a)| / 10^{-7} = |\bar{r}(s,a)| \cdot 10^7$ for all states except for the goal states; thus, the rescaling plots for Reacher (the last figures in each row) suggest that the reshaped reward $|\bar{r}(s,a)|$ is in the range $[0,1]$.

---

> > > > ### Comment · Reviewer_vrFt · 2021-08-20
> > > > **Thanks for the additional analysis**
> > > >
> > > > Thanks very much for the additional explanations and results. The implementation of PBRS looks good to me and the reward scaling plots are helpful.
> > > >
> > > > If I understand correctly, the Reacher results do call into question the hypothesis that "HuRL>PBRS because of reward scaling." HuRL does seem to substantially outperform PBRS on Reacher in the plots that you shared, but the effect of reward scaling is much more dramatic for HuRL than for PBRS in this case. You explained that this is due to the sparsity of the reward. But now I am confused because I thought that HuRL was meant to avoid such dramatic re-scaling? And if the benefit of HuRL over PBRS in sparse reward settings like Reacher can't be explained by reward scaling, what is the explanation? Let me know if I misunderstand.
> > > >
> > > > Thank you for continuing to discuss this point with me. I think it is very important for understanding the contribution of HuRL with respect to PBRS.

---

> > > > > ### Author Response · Authors · 2021-08-21
> > > > > **More clarifications**
> > > > >
> > > > > Happy to help sort this out!
> > > > >
> > > > > Indeed, there was a misunderstanding: our hypothesis from the previous response is that a big factor in PBRS's performance is **erratic short-term** rescaling -- i.e., abrupt rescaling of reward up and down across consecutive iterations, which manifests itself in noticeable jitter in PBRS's rescaling plot lines. This doesn't contradict our earlier statement that PBRS's failure is due to reward rescaling; this is just a refinement of that earlier statement.
> > > > >
> > > > > Our comment regarding Reacher in the previous response was tangential to explaining why PBRS is performing poorly. It was merely aiming to explain why reward rescaling is so large on Reacher, both in case of HuRL and PBRS -- it is because Reacher's reward is sparse, so *any* meaningful reward shaping is bound to induce numerically large rescaling ratio. The plots for Reacher are nonetheless consistent with the hypothesis from our previous reply: on Reacher, like on all other problems we've tried, PBRS introduces significant erratic short-term rescaling changes (significant relative to the absolute variation of PBRS's own rescaling -- see the 6th plot in the bottom row in the doc from the our previous reply) showing as jitter. HuRL doesn't have such jitter relative to the absolute variation of HuRL's own rescaling on Reacher (see the 6th plot in the top row in the doc from the our previous comment), like on all other problems.
> > > > >
> > > > > To drive this point home, note that, conversely, in the very rare cases when HuRL introduces jittery rescaling, this hurts HuRL's learning too. The only such jittery rescaling case we have encountered for HuRL is at the beginning of training for Hopper (the beginning of HuRL's plot in the 2nd figure in the top row) -- look at how it destabilizes HuRL's learning at the beginning (2nd figure in the bottom row).
> > > > >
> > > > > Thus, rescaling jitter can hurt both PBRS and HuRL when it's present, but PBRS is much more prone to inducing it.

---

> > > > > > ### Comment · Reviewer_vrFt · 2021-08-21
> > > > > > **Thanks; score increased**
> > > > > >
> > > > > > Thank you for helping to resolve my misunderstanding. I feel that the contribution of the paper has been strengthened by the additional results and analysis, so I increased my score to 6. I hope that the paper can be updated to include the important points and new experiments from this discussion. Thanks again for taking the time to do the additional experiments and clarify some of the key points.

---

> > > > > > > ### Author Response · Authors · 2021-08-21
> > > > > > > **Thanks!**
> > > > > > >
> > > > > > > We'll add the key points and data from this discussion to the paper and the appendix.

---

### Official Review · Reviewer_58wi · 2021-07-16

**Rating:** 7
**Confidence:** 3

**Summary:**

  This paper proposed an alternative framework for reward shaping in RL
  that uses a heuristic function.

  Unlike potential-based reward shaping, the algorithm does not subtract
  the heursitic value of the current state from the reward.  There is a
  coefficient for the heursitic value to be added to the reward.
  After each episode, it incrementally increases the discounting factor
  (converges to 1) and decreases the coefficient for the heuristic value
  (converges to 0).

  In section 4, the authors theoretically analysed their framework.
  During the analysis, the authors rediscovered *admissibility* &
  *consistency*, concepts that are already well-known in shortest path
  finding literature from the earliest history of AI (can be traced back
  to 1968).  Obviously, being such a fundamental, well-known concepts,
  you are not allowed to rename and replace them, so it is necessary to
  fix these definitions to match the existing literature.  However,
  although these definitions and proofs are not new, I believe *the
  finding that admissibility & consistency will help RL is quite new*,
  and has never been theoretically analyzed before.  This will attract
  the attention of not just the RL community but also of the symbolic
  AI community.


**Limitations And Societal Impact:**

The paper is categorized as basic research rather than applied research. It is not likely to cause large societal nor environmental impact.

**Main Review:**

2 Definition 4.1 and Proposition 4.4 are well-known concepts in shortest path finding literature - rename them
==============================================================================================================

  Despite I am not familiar with the regret & bias analyses in RL,
  definitions and propositions in section 4 are known.  Definition 4.1
  is called *consistency*, and $V^*(s)\geq f(s)$ is called
  *admissibility*.  Proposition 4.4 is known (consistency implies
  admissibility).  Both are important concepts in search algorithm
  literature and in deterministic/stochastic shortest path finding on
  graphs (which is my primary background).

  In shortest path finding problems, we have costs instead of rewards,
  which are typically not discounted, and is aimed to be minimized.
  For a moment, assume a problem of finding the smallest cost path to
  one of the goal states $G$ on a graph with deterministic transitions.
  We define $h^*(s)$ to be the shortest path cost from the current state
  $s$ to the nearest goal $g\in G$.  A heuristic function $h(s)$ is an
  estimate of $h^*(s)$. (see the resemblance to $V(s)$ and $V^*(s)$.)

  An *admissible* heuristic function is an optimistic heuristic function
  that never overestimates $h^*(s)$, i.e., $h(s) \leq h^*(s)$.
  Many heuristic graph search algorithms, most notably $A^*$, are
  guaranteed to find an optimal solution if it uses
  admissible heuristics.

  Moreover, a heuristic function is called *consistent* or *monotonic*
  when, for every action $a$, $h(s) \leq c(s,a,s')+h(s')$.  Obviously,
  this is a deterministic Bellman equasion.  With a consistent
  heuristics, $A^*$ has a more desirable guarantee that it never
  re-expands the same node.  The name derives from the fact that the
  heuristic value monotonically decreases toward the goal on any optimal
  solution, and thus consistently guides the search toward the goal
  (without misguiding along the path; i.e., without increasing values).
  Consistency implies admissibility.

  These fundamental facts are discovered by the seminal paper of $A^*$
  by Hart et al, 1968.  You can find them in every symbolic AI textbook
  that most surely covers $A^*$, such as [1], but I guess you are only
  familiar with post-deep-RL literature and lack history.  Later, these
  concepts are extended to stochastic domains by [2] when they describe
  Loop-AO* algorithm for stochastic shortest path problems.

  [1] Stuart J. Russell, Peter Norvig: Artificial Intelligence: A Modern
  Approach (4th Edition). Pearson 2020, ISBN 9780134610993

  [2] Hansen, Eric A., and Shlomo Zilberstein. "LAO∗: A heuristic search
  algorithm that finds solutions with loops." Artificial Intelligence
  129.1-2 (2001): 35-62.


3 Other comments
================

  Since I am not familiar with the theoretical analyses of RL
  algorithms, it is difficult for me to understand Section 4.  So I
  instead have several high-level questions to the authors that could be
  perhaps implied by the theorem but is not communicated effectively to
  the wider audience.

  In l.350, you mention "our reshaped MDP can be shown to be equivalent
  to ... PBRS". Where is this shown? Why this is true?

  I am not quite sure if the proposed HuRL guarantees that the resulting
  policy will converge to the optimal in the original MDP.  Otherwise,
  unlike potential-based reward shaping, this is like shifting the goal
  post, and it is theoretically less appealing because it is not
  optimizing what we want to optimize.


  This paper does not specify the exact definition of f(s).  Is it an
  estimate of V*? If it is an estimate of V*, is it an estimate of the
  cumulative discounted reward with the same discount factor?
  For heuristic functions in graph search, these questions do not arise
  because we do not typically use discounting.


        However, PBRS does not fully leverage information provided
        by the heuristic because it still requires the learning
        agent to solve a long horizon problem, which is the main
        source of difficulty in RL. Conceptually ∗when the
        heuristic is the optimal value function f = V , the agent
        should be able to find the optimal policy π∗ of M by
        acting myopically, as V ∗ already contains all necessary
        long-term information for good decision making. However RL
        with PBRS does not enable such a solution concept.

  I believe this is a miunderstanding.  The authors does not
  sufficiently explain why "PBRS does not enable such a solution
  concept", And indeed the proposition "PBRS does not enable such a
  solution concept" is incorrect.  ~r(s,a,s') = r(s,a,s') + γh(s')-h(s)
  is equivalent to V*(s) = ~V*(s) + h(s) and Q*(s,a) = ~Q*(s,a) +
  h(s). [Ng, Harada, Russel, 1999] So if you just adjust the value of
  ~V* by h post-hoc, then acting myopically according to ~V*+h is
  equivalent to acting myopically according to V* (Same for Q*.)


        In the planning literature, this is typically achieved by
        relaxing the problem using domain knowledge.  ...
        Although using heuristics to accelerate search has been
        popular in planning and control, such as $A^*$ [12], MCTS
        [13], and MPC [7, 14–16], its theory is less developed for
        the setting where the MDP is unknown.

  Actually, [i,ii] learns a world model which is compatible with
  planning formalisms and can directly benefit from relaxation-based
  heuristics and shows substantial speedup without any policy learning.

  [i] Masataro Asai, Hiroshi Kajino, Alex Fukunaga, Christian Muise:
  Classical Planning in Deep Latent Space. CoRR abs/2107.00110 (2021)
  [ii] Masataro Asai, Christian Muise: Learning Neural-Symbolic
  Descriptive Planning Models via Cube-Space Priors: The Voyage Home (to
  STRIPS). IJCAI 2020: 2676-2682


  Finally, the evaluation does not include comparisons against
  potential-based reward shaping.  HuRL and PBRS are given the same
  amount of information from the environment --- if you claim HuRL is
  better than PBRS, you should definitely compare the results.


**Time Spent Reviewing:**

4

---

> ### Author Response · Authors · 2021-08-11
> **Response to reviewer #58wi**
>
> ### **Admissibility and consistency**
>
> We would like to respectfully assure you that we have heard about these concepts once or twice before ;)  But our concepts of “pessimism” and “improvability” are *not* the same as “admissibility” and “consistency”, respectively. They are actually the opposite, in some sense.
>
> Namely, admissibility is a formal notion of heuristic *optimism*. The definitions you cited capture this notion in the *cost minimization* setting, where being admissible means *underestimating the cost* (of getting to the goal). In our *reward maximization* setting, being admissible means *overestimating* (expected discounted long-term) *reward*  that can be collected starting from a state. But our theory desires heuristics that *underestimate reward*, i.e., capture the notion of *pessimism* (see Section 4.4)
>
> We could call this notion “strict inadmissibility”, but this is cumbersome and, more importantly, the term “pessimism” applied to (Q-) value functions is already firmly established in RL.
>
> Similarly, “improvability” is not the same as “consistency”. While the latter implies admissibility, the former implies strict inadmissibility/pessimism (see Prop 4.4 with $\lambda=1$), which is what we need for our theory.
>
> Thus, loosely speaking, our theory says -- counterintuitively, from the standpoint of planning -- that *inadmissibility* of a heuristic is desirable. However, from the standpoint of certain subareas of RL, especially offline RL, this result isn’t entirely surprising. Note the contrast with planning algorithms such as LAO*. There, an agent doesn’t explore randomly: it is always greedy w.r.t. the current value function, and heuristic admissibility ensures that greedy action selection always leads the agent to discover the optimal policy. In RL algorithms, on the other hand, an agent can discover an optimal policy independently of whether its heuristic is admissible or not, but at the cost of doing (non-greedy) exploration.
>
> Among all pessimistic heuristics, our theory suggests that improvable heuristics are particularly favorable, because the performance bias of learning with such heuristics depend only locally on how good the heuristic approximate V* on the paths the optimal policy would visit, rather than globally (see Proposition 4.3).
>
> **All that said, we will discuss these connections to admissibility and consistency in the final paper version to help readers connect pessimism and improvability to these familiar concepts and avoid confusion.**
>
>
> ### **Comparison to PBRS**
>
> We have performed such a comparison and discussed in detail how HuRL is preferred over PBRS; please see the **Common response to all reviewers** comment.
>
>
> ### **Other comments**
>
> **Convergence to the optimum in the original MDP.** Learning with HuRL would converge to the original optimal policy because eventually $\lambda_n$ becomes 1 and the agent ends up facing the original MDP without the effect of heuristics. This can be seen by plugging in $\lambda_{\infty}=1$ in Eq. 2. In practice, the convergence can happen sooner, before $\lambda_n$ actually reaches 1, because of the known Blackwell optimality property [22].
>
> **l.350: where is the proof?** Please see the **MDP equivalence between HuRL and PBRS.** section of the **Common response to all reviewers** comment.
>
> **Is $f(s)$  an estimate of $V^\*(s)$?** Yes, $f(s)$ can be thought of that way: in Proposition 4.2 (line 227), we assume $f(s) \in  [0, 1/ 1−\gamma ]$, and on line 85 we assume that (discounted) value functions $V$ are in the same range. Admittedly, this is difficult to piece together based on the current presentation. We will update the paper to make this more explicit.
>
> **Why does PBRS not fully leverage information provided by the heuristic?** Please see the **PBRS vs. HuRL for policy *learning*.**  section of the **Common response to all reviewers** comment. Does this clarify the issue?
>
> **Making use of relaxation-based heuristics by learning a compatible MDP representation as in [i, ii]**. This is relevant indeed! We will mention them in the paper as model-based ways of using heuristics to solve initially unknown MDPs whose structure is expressible in STRIPS.
>
> **Empirical comparison against PBRS.** We now realize that this was a significant omission and have performed such a comparison. Please see the description and results in the  **Empirical comparison of HuRL and PBRS.** section of the **Common response to all reviewers** comment. We will update the paper to include them.

---

> > ### Comment · Reviewer_58wi · 2021-08-11
> > **Thanks for the replies**
> >
> > The reply has solved many of the concerns I had, which I am happy with. I will increase the score conditioned by that the authors will revise the paper as in the reply.
> > Further comments:
> >
> > * Re: Admissibility and Consistency:
> >
> > I see that Admissibility and Consistency are about optimism (underestimation for cost minimization, overestimation for reward maximization)
> > while your claim is about pessimism (overestimation for cost minimization, underestimation for reward maximization).
> >
> > Perhaps this may have to do with the fact that RL and MDP are more similar to all-pair shortest path with possibily negative costs, like in Floyd–Warshall Algorithm (Bellman-Ford for single shortest path, Jonsson for all-pair shortest path with no cycle), rather than a single shortest path finding with positive cost, like in Dijkstra algorithm.
> > You could survey the relevant literature to see if equivalent concepts already exist for Floyd-Warshall.
> > Connections could also be found from longest-path finding algorithms.
> > Importantly, since research is not about flag painting but about making a connection to the past, I appreciate authors' promise on the future revision.
> >
> > Another thing that could be possibly considered is that perhaps *both* admissible heuristics *and* pessimistic heuristics could help the training. I understand that pessimistic heuristic would help the training, but I believe it did not say the admissible heuristics does not help the training (can you comment on this?). There are two reasons I felt so:
> > * A vast majority of heuristics being used in the past, such as Euclidian distance in 2D pathfinding, are admissible heuristics.
> > * In Pearl's textbook _Heuristics_ (1984) and the paper it based on [1], he discussed the desirable property of inadmissible heuristics (which is still not well analysed in the search community sadly!). He showed that it is more important for the inadmissible heuristics to be precise (variance) than being accurate (bias), which is essentially the characteristics about both the lower (optimistic) and upper (pessimistic) bound of the heuristic estimate.
> >
> > [1] Pearl, Judea. "Heuristic Search Theory: Survey of Recent Results." IJCAI. Vol. 1. 1981.
> >
> > * $f(s)\in[0,1/1-\gamma]$
> >
> > I see, so this corresponds to $f(s)\in[0,\infty]$ in an undiscounted case. Since most easily implementable heuristics are considering an undiscounted case (e.g., Euclidian distance is an undiscounted sum of relaxed path cost = straight line), you should note that they must be transformed appropriately.

---

> > > ### Comment · Reviewer_58wi · 2021-08-11
> > > **minor comment**
> > >
> > > Heuristics are conventionally denoted by $h(s)$, so consider changing the notation.

---

> > > ### Author Response · Authors · 2021-08-12
> > > **A few more words about heuristic (in)admissibility in HuRL and in heuristic search**
> > >
> > > We are glad that our response has helped address many of your questions and concerns!
> > >
> > > Regarding heuristic (in)admissibility, we have thought more about connections to various search settings, including settings with negative costs, but so far have concluded that its role (at least its role as analyzed in *most of* the search literature) is just very different in search versus in HuRL. Most of the search literature considers admissibility as desirable because it prevents eliminating the optimal solution from consideration. In HuRL though, no heuristic -- whether admissible or inadmissible -- has the "power" to eliminate the optimal policy from consideration: when $\lambda_n \rightarrow 0$, the HuRL agent eventually ends up facing the original MDP with no heuristic, where its RL algorithm (at least, vanilla tabular RL in discrete-state and -action settings) can still discover the optimal policy. However, in HuRL the heuristic can facilitate learning the optimal policy quicker by helping the agent discover decent policies by the time the effect of the heuristic vanishes as above.
> > >
> > > I.e., the role of heuristic (in)admissibility in search has been considered mostly from the standpoint of solution optimality, whereas HuRL considers it from the standpoint of the amount of computation/sample complexity.
> > >
> > > Could admissible heuristics help HuRL? HuRL can certainly work with any heuristics that approximate $V^*$ well (Corollary 4.1) and is not limited to the improvable ones. So, we expect certain admissible heuristics would help too, though not all (e.g. f(s) = 1/1-gamma is admissible but it is not useful in learning). That said, currently out theory has a more precise characterization of how heuristics help for the  strictly inadmissible rather than admissible case (Proposition 4.3).
> > >
> > > Further investigation into the role of admissible heuristics for RL is an interesting future direction, especially in light of this work that you mentioned:
> > >
> > > > [1] Pearl, Judea. "Heuristic Search Theory: Survey of Recent Results." IJCAI. Vol. 1. 1981.
> > >
> > > Thanks for bringing it up, we'll point out in the revised paper version! This line of thinking looks especially promising than from the standpoint of potentially linking heuristics theory for search and for RL because it pays attention at the connection between heuristics' properties and the amount of search computation they induce, which is also the perspective HuRL takes.
> > >
> > > Regarding heuristic notation, we'll switch to $h(s)$ to make it less cryptic, and point out that heuristics designed for the undiscounted case need to be adjusted for the discounted-reward case.

---

> > > > ### Comment · Reviewer_58wi · 2021-08-16
> > > > **Thanks**
> > > >
> > > > Thanks for giving the interesting thoughts.
> > > >
> > > > I was keep thinking about an explanation of this, and I think it is something to do with dense & sparse reward.
> > > >
> > > > > its role (at least its role as analyzed in most of the search literature) is just very different in search versus in HuRL.
> > > >
> > > > I agree. However, there is a misunderstanding in your comment:
> > > >
> > > > > I.e., the role of heuristic (in)admissibility in search has been considered mostly from the standpoint of solution optimality
> > > >
> > > > Satisficing planning is one of the major track in International Planning Competition. About a half of the papers are about satisficing planning, and most practical applications uses satisficing planning, rather than optimal planning.
> > > > I believe the reason optimal planning seems dominant to the outsiders is that satisficing planning algorithms are more heuristic (i.e., empirical) and have less theory, and thus not much can be taught in grad schools or textbooks. (Personally I wish to change this situation...)
> > > >
> > > > In a satisficing (non-optimal, approximated) setting of search problems, one can use inadmissible heuristics. Neither do we use optimising algorithms, like A*, but rather its greedier variants, e.g., GBFS or Weighted A*. In IPC competitions, iterative WA* is used (See LAMA planner in classical planning). This starts the search with GBFS (equivalent to WA* with weight treated as infinity), then incrementally decreases the weight in Weighted A*, where the last iteration is weight = 1, which is equivalent to A*. It finds a satisficing solution quickly, then improves the solution (finding a solution with less costs). The solution cost of the plan found in the current iteration becomes the solution **upper bound** of the next iteration, thus any node which exceeds this bound will be pruned.
> > > >
> > > > I feel this satisficing setting is more similar to the RL setting. Recall that Mujoco tasks are dense reward domains --- Being dense reward, getting a reward is easy, while traditional search problems have mostly negative rewards (costs) with sparse positive rewards. In dense reward task, finding a satisficing solution is trivial -- any action will give some reward, so any sequence of actions is a solution. So the goal is to find a better solution with a larger reward. **The search version** of this is to find a satisficing solution first (which is trivial), then minimize the cost.
> > > >
> > > > This gives a simple explanation of why pessimistic heuristics works --- RL is using heuristics mostly as an upper bound of the cost, rather than lower bound, similar to iterative WA*. Because it handles dense reward domains, where finding the first solution is easy, most of the effort is spent on reducing the cost (maximizing the reward).
> > > >
> > > > Therefore, there is a chance that the ratio of positive / negative rewards will affect the efficacy of admissible (optimistic) and pessimistic heuristics.

---

> > > > > ### Author Response · Authors · 2021-08-20
> > > > > **Re:heuristic optimism/pessimism <-> reward density**
> > > > >
> > > > > Right, when we were making the comment about in/admissibility in search having been considered mostly from the optimality standpoint, the “mostly” applied only to the theoretical search literature, since our own study of heuristics in RL is of a more theoretical flavor. We are aware that the empirical planning literature, including stochastic MDP planning literature, has studied and used inadmissible heuristics extensively, and didn’t mean to imply otherwise.
> > > > >
> > > > > Regarding the connection between the usefulness of in/admissible heuristics and reward density, such a connection is certainly possible and is indeed an interesting avenue to explore! Note that our existing theoretical arguments in favor of heuristic pessimism for HuRL don't make any assumptions about reward density, so they are valid both for "typical" reward structures considered in planning and for the "typical" ones considered in RL. A plausible extension of our theory to the special cases of, e.g., goal-directed sparse-reward problems (by the way, one of the environments in our experiments, Reacher, is of this type) may be able to identify conditions and reasons that make *optimistic*/admissible heuristics desirable for HuRL as well.

---

### Official Review · Reviewer_RNky · 2021-07-17

**Rating:** 7
**Confidence:** 3

**Summary:**

This work presents a theoretical framework for reasoning about solving MDPs in which agents have access to pretrained "heuristics", which are defined as functions that provide some prior over the optimal value function. The framework is used to define a regularization technique for mixing a such heuristics into the training of a standard model-free RL pipeline, presenting experiments on Mujoco and Procgen with a variety of heuristic baselines.

**Limitations And Societal Impact:**

Some limitations of the work were discussed, however I have raised some points in the main review about adding some further discussion and clarity on some of them.

I haven't seen an explicit discussion on societal impact of the work in the manuscript.

**Main Review:**

TL;DR -- This is a relatively good paper. The work presents a sound regularization technique for incorporating some prior knowledge about the task, as well as a good theoretical framework for reasoning about the method and potential follow ups. There remain some doubts / questions about the breadth of classes of heuristics that can be effectively used in this setting.

I enjoyed reading the manuscript. It is well written, mostly backed up by a good literature research, and has a decent amount of low level details in both the main document and the appendix.

Things that I think could be improved:

1. I think "heuristics" here is used a little too across the text. Before getting to Section 2, I was under the impression that the method would have involved some form of heuristic *policy*, rather than a "prior guess of the optimal value function $V*$" (Section 2.2). Other readers might be confused by this mismatch, and I think it'd be appropriate to think about a different nomenclature. If anything, is there prior work that uses the same definition?

2. Much of the presented experimental setting relies on fitting a Monte-Carlo regressor on some observed data. While sound, it seems quite a relatively naive / weak heuristics whose effectiveness is highly dependent on the shape and behavior of task / reward function. In particular, in the experiments the manuscript does not seem to explicitly declare how the data is collected. Are the various heuristics trained from data collected by SAC at different training times? If so, how do they qualitatively look like wrt. the presented tasks?

3. It's a little hard placing this work in the existing literature because there's a vast amount of work that tries to utilise prior information about the task (e.g. via knowledge distillation, transfer learning, expert-based training) in all sorts of contexts. Some of this work is covered in the manuscript's lit review, but there's much that isn't:
  a. https://ojs.aaai.org/index.php/AAAI/article/view/5963
  b. https://arxiv.org/abs/2009.07888
  c. https://arxiv.org/abs/1905.01240
  d. https://arxiv.org/abs/1905.06750 (and so forth...)

 t think a lot of this stems from the paper pushing an overly general narrative about "heuristics", while actually using a rather narrow definition and formalisation of it.

So, ultimately I am going to recommend for acceptance, but I would like some additional clarity on the following to improve my score:

- How's this method generally superior to standard distillation methods in RL? (Or at the very least how does it compare generally?)
- Considering many heuristics out in the wild are in the form of more or less fixed policies, what could be some strategies for adapting these to be employed in HuRL? (It's probably okay to hypothesise, guess, or even present naive / baseline ideas, but at the very least discussing this in a informed manner in the manuscript would be useful)
- How do the heuristics used in the experiments generally look like? (i.e. generally more clarity on this, rather than handwaveing them as just monte-carlo regression of some data)

## After-rebuttal Edit

Given the authors' responses to my and other reviewers comments, I am happy to raise my score to full accept.

**Time Spent Reviewing:**

5

---

> ### Author Response · Authors · 2021-08-11
> **Response to reviewer #RNky**
>
> ### **Definition of the term “heuristic"**
>
> We didn’t mean to suggest that the interpretation of the term “heuristic” that we use in this paper is universal. Interpreting “heuristic” as a heuristic policy is perfectly valid and common as well. While we tried to highlight early on in line 42 in Sec 1 that a heuristic in our work represents a prior guess of V*, in the final version, we will be much more explicit early on both about the fact that in our work heuristics are candidate value functions and about the fact that, more broadly, there are other definitions of a heuristic.
>
> That said, interpreting heuristics as functions is indeed very standard.This originally comes from deterministic planning (e.g., the textbooks by Russell and Norvig) and was adopted in the MDP literature as well. See, e.g., Mausam and A. Kolobov’s “Planning with Markov Decision Processes: an AI Perspective” textbook (unfortunately, we can’t find it in open access, but many institutions have an electronic copy), whose Definition 4.4 describes a heuristic as a “a value function that provides state values when an MDP solution algorithm inspects them for the first time”.
>
> ### **Deriving heuristics from data via MC regression and what they look like**
>
> We agree that in general, deriving heuristics by MC regression from arbitrary data is a dubious idea. In practice, however, the data from which we envision deriving heuristics in this way isn’t arbitrary. In many settings, from computer systems to customer relationship management, the RL agent has access to data gathered by a hand-crafted, possibly rule-based policy, but the policy itself is not available. For example, in customer relationship management, this trajectory data was generated by human personnel interacting with customers by loosely following some guidelines. MC regression on such data is very meaningful -- qualitatively, it approximates the optimal values of states by the values according to the policy that generated the training data.
>
> This is exactly the setting we model in our experiments, which we describe in the 1st paragraph of Section 5.1. We  provide the missing details of how we trained the heuristics used in the experiments in the **How the heuristics were generated via MC regression** section of the **Common response to all reviewers** comment. Namely, as you correctly guessed, we collect the data for MC regression by training by taking SAC policy snapshots during different training times. We then ran the resulting policies to generate trajectory data, and ran MC regression on a mix of trajectories from all such policies.
>
> At a high level, these heuristics approximate $f(s) \approx \frac{1}{K} \sum_{k=1}^K V^{\mu_k}(s)$, where $\mu_k$ is the $k$-th behavior policy. As these behavior policies might have different state coverage, the heuristic would behave like $V^{\mu_k}(s)$ if only $\mu_k$ visits the state $s$, not the others, for example.
>
> The best policies in these mixtures were still quite poor, barely able to perform the target task. E.g., in the Humanoid environment, the best of them enabled the humanoid to walk without falling but very slowly.
>
>
> ### **Positioning in the context of other work, esp. distillation, transfer learning, expert-based training, etc**
>
> HuRL indeed belongs to the broad class of methods that make use of prior knowledge in learning, as do the methods you listed. However, there are two major differences:
>
> -	**The form of prior information that these works expect.** In transfer learning and distillation, it usually comes in the form of features,  and in imitation learning -- in the form of an expert, possibly an interactive one, whose performance the learning agent attempts to match. In HuRL’s case, this information is expected in the form of state value estimates. The differences in assumptions about prior data make techniques from these literature areas vastly distinct.
>
> - **During learning, a HuRL agent doesn’t aim to internalize any of the prior knowledge.** It uses it to guide its learning from its own environment interactions, and the effects of this knowledge are eventually removed by HuRL completely, when $\lambda_n \to 1$.
>
> This is an excellent question though and made us think a bit. We’ll mention this distinction in a revised paper version.
>
> ### **Incorporating heuristic policies in HuRL**
>
> We think that a direct way of leveraging a heuristic policy in HuRL is to use it as the initial policy in policy optimization, and use its value function (which e.g. can be estimated by MC regression with the samples collected by the heuristic policy). When there are multiple heuristic policies, the policy initialization can be set as the best performing one among them, and the heuristic can be set as the average value function (as we did in the experiments) or by max-aggregation in [44]. In summary, if we have access to a heuristic policy, then besides using this policy for warm-starting HuRL will benefit from using this policy’s value function as a heuristic, and MC regression is one way to learn this value function.

---

> > ### Comment · Reviewer_RNky · 2021-08-27
> > **Thank you for the response**
> >
> > Thank you for the thoughtful response. Generally I am happy with your arguments (and with your response to the other reviewers' questions), so I'll be increasing my score to 7.
> >
> > >Definition of the term "heuristic"
> >
> > I think that's a fair point. I do generally advise to be conservative (or at the very least extremely clear) with regards to employing terms that have different semantics across different sub-communities, but in this case it seems like it'd be pretty straightforward to rely on the reference as way to clarify the meaning (or -- rather -- the implementation) of the term heuristics. Please make sure that the reader's expectations are set in a reasonable manner while reading the abstract / introduction.
> >
> > >Deriving heuristics from data via MC regression and what they look like
> >
> > Thanks, I do agree that MC regression makes sense when used in this manner. Please make sure some of this discussion is included at the very least in the appendix, as I imagine future readers might have similar questions.
> >
> > >This is an excellent question though and made us think a bit. We’ll mention this distinction in a revised paper version.
> >
> > Thank you, I'm glad I was of help :) Sadly we'll need to wait for the camera-ready version to see the final discussion in the manuscript, but I'm relatively confident the writing will be improved, given the clarity in the rebuttal.

---

> > > ### Comment · Reviewer_58wi · 2021-08-27
> > > **Regarding "heuristic"**
> > >
> > > Just to note, in application areas (e.g. mechanical enginnering) or in black-box optimization community, the word "heuristic" can be sometimes automatically interpreted as metaheuristic methods such as Genetic Algorithms / Particle Swarm Optimization / tabu search. Another reason to be careful.

---

> > > ### Author Response · Authors · 2021-08-28
> > > **Thanks!**
> > >
> > > We are glad our response has helped clear the ambiguities, and will transfer over the explanations to the final paper version. Regarding your and **Reviewer 58wi**'s points about the term "heuristic", indeed we've realized as a result of the discussions here that our intended interpretation of this term is far less universal than we originally thought. We will explicitly state early on in the paper that in this work we are focusing on the "value function" interpretation of it, and briefly state how it is related to the other interpretations in Related Work.

---

### Official Review · Reviewer_bqji · 2021-07-20

**Rating:** 5
**Confidence:** 3

**Summary:**

The paper aims to solve/mitigate the long term credit assignment problem in reinforcement learning. It does so by pre-training heuristic value functions and using them to shorten the horizon. The claim is that the proposed approach could potentially improve sample complexity, like other warm-start or regularization methods in RL.

**Limitations And Societal Impact:**

The authors should address the impact of using biased data (and the quality of the dataset in general) for pre-training the heuristic value functions

**Main Review:**

The authors proposed a novel variant of the existing ideas (of warm starts) and gave some theoretical reasoning. However, the experiments are very limited in nature:
1) The idea was proposed as a solution to the long-term credit assignment and long horizon problems in reinforcement learning but the results were demonstrated on simple MuJoCo tasks
2) The details of how long (number of time steps or episodes?) the value function was trained is unclear. A comparison of a standard algorithm (like SAC) using this pre-trained value function vs HuRL-MC would be useful.
3) Am I understanding correctly that in the plots, HuRL-MC starts with pre-trained value function whereas SAC starts training from scratch? I do not think this is a fair comparison
4) The authors cited several other warm-starts and regularization based approaches but no performance comparison with those algorithms was provided thus making it difficult to understand the effectiveness of the proposed algorithm
5) The authors mentioned that for each environment, they performed hyper parameter tuning for learning rates and the discount factor of SAC and then $\lambda_0$ and $w$ were tuned for each heuristic of each environment. In my opinion, this is an extreme overkill and consumes a lot of computational resources. I recommend authors to clarify this and provide comparisons of the computational resources used (including resources for hyper parameter tuning)

Overall, the proposed algorithm is a novel variant and the authors provided sufficient theoretical backing but the experiments are seriously lacking. The current experimental results do not demonstrate the superiority of the algorithm at all. I am willing to increase my score if my 5 concerns above are addressed. Additionally, it would be nicer to relate the proposed algorithm with the curriculum learning literature as the setup seems very similar.

**Time Spent Reviewing:**

6 hours

---

> ### Author Response · Authors · 2021-08-11
> **Response to reviewer #bqji**
>
> ### **Clarification regarding warmstarting.**
>
> We would like to clarify that by the term “warmstarting” in this paper we don’t mean methods for *computing* heuristic value functions -- these are neither a contribution nor the focus of this paper. In fact, as we illustrate by additional experiments in the **HuRL with Heuristics based on Domain Knowledge** section of the **Common response to all reviewers** comment, we heuristics may well be handcrafted.  Instead, the main claimed novelty of this work is a method for using heuristics in RL (HuRL)..
>
> We see several places in the paper that could have led to confusion about this point. Our apologies, we will fix them.
>
>
> ### **1. Experiments on more complex task.**
>
> We are happy to consider additional  tasks for our experiments, in case you have specific suggestions. In the meantime, we have conducted an additional experiment on a modified Reacher-v2 task -- its sparse reward makes it very hard for SAC, but HuRL easily solves it with a heuristic. Please see the **HuRL with Heuristics based on Domain Knowledge** section of the **Common response to all reviewers** comment for the results.
>
>
> ### **2. Details of how the heuristics were generated.**
>
> Please see **How the heuristics were generated via MC regression** section of the **Common response to all reviewers** comment.
>
> Because the Garage’s implementation of SAC only has a Q function, there is no clear way to initialize the state-action Q function in SAC with a value function (i.e. a function of just state). We now realize that there are other implementations of SAC that do use a state value function as well, which might be a source of confusion.
>
>
>
> ### **3. Did HuRL use a pre-trained value function in the experiments?**
>
> No algorithm in the experiments (including HuRL variants) used pretrained value functions. We will clarify this in the revision. Since each algorithm in the experiments sees the same heuristic, initial policy, and initial value function, we think that the current experiment protocol gives a fair comparison for the setup that we wish to study in the paper (i.e. online RL with a given heuristic).
>
>
> ### **4. Other warm-start techniques.**
>
> The experiments currently in the paper provide a comparison to the most common warm-starting technique -- BC (behavior cloning). They show SAC with/without BC as well as SAC+HuRL with BC. Since SAC with BC and SAC+HuRL with BC use the same initialization of policies and same random initialization of value functions, this is a fair comparison that shows the benefit of HuRL.
>
> The other major warmstarting technique we mentioned in the paper is potential-based reward shaping (PBRS). The submission indeed didn’t have experiments with PBRS, but we have since realized this was a major omission, and in the **Empirical comparison of HuRL and PBRS** subsection of the **Common response to all reviewers** comment, have added new experiments on SAC+HuRL vs. SAC+PBRS using the same heuristics. The results show that HuRL can better leverage the heuristic to achieve more efficient learning.
>
>
>
> ### **5. Hyperparameter Tuning.**
>
> In practice HuRL doesn’t need the extent of hyperparameter tuning we did for the experiments -- we did it only for a cleaner empirical comparison. Please see the  **HuRL and hyperparameter selection** section of the **Common response to all reviewers** comment.
>
>
> ### **Relationship to curriculum learning**
>
> We will elaborate the discussion (currently only hinted at in Lines 62, 114 and 395) relating curriculum learning techniques to HuRL in the revision. The most related approach is Atari-Reset https://openai.com/blog/learning-montezumas-revenge-from-a-single-demonstration/ and Reverse Curriculum in Robotics https://bair.berkeley.edu/blog/2017/12/20/reverse-curriculum/ where an RL agent is initialized to states drawn from an expert demonstration in goal-oriented tasks. Their curriculum creates a short horizon RL problem by initializing the agent close to the goal, and when the agent has successfully learned to reach the goal, the agent is initialized further away. This gradual increase in the problem horizon inspires HuRL’s approach to increasing $\lambda$. However, unlike these methods, HuRL does not require the agent to be initialized on expert states and can work with many different base RL algorithms and provides strong theoretical guarantees.
>
> =================
>
> **UPDATE**: We realized we forgot to comment on your point regarding the impact of using biased data for pre-training heuristic value functions when posting the original response:
>
> ### **The impact of using biased data for pre-training heuristic value functions**
>
> The bias in and the general quality of the data used for training the heuristics matters only indirectly, via the properties of the heuristic produced from it, such as heuristic improvability and approximation quality w.r.t. $V^*$. How the quality of data translates to these properties highly depends on the algorithm for training a heuristic: MC regression can give different results than VAE-based or other methods one may choose.
>
> Note also that, as illustrated in the experiment in the **HuRL with Heuristics based on Domain Knowledge** subsection of our **Common response to all reviewers** comment, effective heuristics don't even need to be trained from data.
>
> We will mention this in the revised paper version.

---

### Author Response · Authors · 2021-08-11
******Common response to all reviewers******

We thank all the reviewers for their constructive and detailed feedback. We have conducted empirical comparisons to their suggested baselines, included discussions on the relationship between HuRL, PBRS and related (but surprisingly different) concept of heuristic admissibility in search, and have clarified terminology throughout the paper. We believe that addressing the reviewers’ suggestions has substantially improved the paper. In this comment we answer issues raised by at least 2 reviewers, and respond to individual concerns in individual responses for each review.


### **Relationship to PBRS ----- (Reviewers bqji, vrFt, 58wi)**

**PBRS vs. HuRL for policy *learning*.**  PBRS can use any heuristic function $f$ of the state to reshape the reward, while preserving the ordering of policies. When the heuristic is the optimal value function., i.e., $f= V^*$, acting greedily with the PBRS reward achieves the optimal performance. However, transforming a reward with **PBRS** and then giving it to an RL algorithm **does not necessarily lead to faster *learning*** than with the original reward function. Our framework **HuRL is designed to address this issue**, so that common RL algorithms can leverage the short-horizon potential provided by the PBRS structure to achieve faster learning.

More precisely, PBRS only enables the *possibility* of designing an efficient RL algorithm. But even when the heuristic in PBRS is exactly $V^*$, running a typical RL algorithm based on Value Iteration or Policy Iteration (like SAC) with the PBRS reward is not always more efficient, as the RL algorithm does not *know* that acting greedily with respect to the reshaped reward is a good thing to do and may seek to explore further for resolving long-term credit assignment.



**MDP equivalence between HuRL and PBRS.** The way HuRL helps policy *learning* is that, as mentioned in Section 6 (line 352), it runs a base RL algorithm with *both* the PBRS reward and a smaller guidance discount, which encourages the base RL algorithm to act more myopically.

This connection can be seen from the following derivation: In Eq. (1), HuRL uses an MDP with a reward $r(s,a) + (1-\lambda)\gamma f(s’)$ and a small guidance discount $\lambda\gamma$.  By PBRS theory, solving this MDP is equivalent to solving another  $\lambda\gamma$-discounted MDP whose reward is $r(s,a) + (1-\lambda) \gamma f(s’) +  \lambda\gamma f(s) - f(s) = r(s,a) +  \gamma f(s’) - f(s)$, which is exactly the definition of PBRS reward. That is, HuRL’s and PBRS MDPs are the same up to PBRS transformation.



**Empirical comparison of HuRL and PBRS.** In this rebuttal, we include SAC with the PBRS-transformed reward in the Mujoco environments already used in the original submission (30 seeds), as requested by some of the reviewers. This extra SAC+PBRS baseline uses the same heuristic as SAC+HuRL but transforms the reward in the standard PBRS fashion and does not further lower the discount. In comparison with SAC w/ BC, the only difference is that SAC+PBRS uses the PBRS reward in learning. We thank the reviewers for pointing out this important baseline.

The experimental results can be found in the anonymous link:

https://figshare.com/s/d5b21f8a7aca565df8da

We see that applying PBRS to SAC leads to even worse performance than running SAC with the original reward let alone SAC+HuRL, agreeing with the hypothesis in section **PBRS vs. HuRL for policy *learning*.** above.

There are two reasons why SAC+PBRS is less desirable than SAC+HuRL: (1) PBRS changes the reward/value scales in the induced MDP, and popular RL algorithms like SAC are very sensitive to such changes. In contrast, HURL induces values on the same scale as we show in Proposition 4.2. (2) In HURL, we are effectively providing the algorithm some more side-information (i.e. $\lambda$) to let SAC shorten the horizon when the heuristic is good.


### **HuRL with Heuristics based on Domain Knowledge ----- (Reviewers bqji, vrFt, 58wi, RNky)**

We wish to highlight that HuRL does not assume the heuristic is learned from offline data. Following the suggestions of **Reviewer vrFt**, we conducted a goal-oriented experiment with sparse reward. The environment is based on Reacher-v2 in Mujoco, but we change the reward to be 0 when the robot’s end effector is within the goal’s 0.01 radius, and -1 otherwise. The description of the heuristic $f$ and the  experimental results (30 seeds) can be found atthe anonymous link:

https://figshare.com/s/d5b21f8a7aca565df8da

The experiment was done following the same protocol used in other experiments; the baseline SAC was fine tuned for this environment, and SAC+HuRL used the hyperparameters from baseline SAC and an additionally tuned $\lambda$ schedule. The policies and value functions were randomly initialized. We see that in this sparse reward experiment SAC and SAC+PBRS struggle to learn, while SAC+HuRL is able to converge to the optimal performance much faster.

We will add these results to a revised version of the paper.


### **How the heuristics were generated via MC regression ----- (Reviewers bqji, RNky)**

We designed this heuristic generation experiment to simulate the typical scenario where offline data collected by multiple policies of various qualities is available before learning. In this case, a common method for inferring what values a good policy could get is to inspect the realized accumulated rewards in the dataset.

Specifically, for each mujoco experiment, we ran SAC for 200 iterations and logged the intermediate policies for every 4 iterations, resulting in a total of 50 behavior policies. In each random seed of the experiment, we performed the following: We used each behavior policy to collect trajectories of at most 10,000 transition tuples, which gave about 500,000 offline data points over these 50 policies. These data were used to construct the Monte-Carlo regression data, which was done by computing the accumulated discounted rewards along sampled trajectories. Then we generated the heuristic used in the experiment by fitting a fully connected NN with (256,256)-hidden layers using default ADAM with step size 0.001 and minibatch size 128 for 30 epochs over this randomly generated dataset of 50 behavior policies. The same dataset was used to initialize policies with BC if applicable.

We apologize that we did not provide enough experimental details in the submission. We will add the missing information in the revision.


### **HuRL and hyperparameter selection ----- (Reviewers bqji, vrFt)**

We chose this hyperparameter tuning procedure to make sure that the baselines (i.e. SAC) compared in these experiments are their best versions. For HuRL, we only tuned $\lambda_0$ and $w$, without changing those hyperparameters already in SAC.

While this procedure indeed takes more compute (the computation resources used are reported in Appendix C.1; the tuning of the base SAC takes up most of them), we performed it to produce more reliable empirical results and avoid making a false claims due to luckily (or unluckily) using some hand-specified hyperparameters or a particular way of aggregating scores during tuning hyperparameters across environments.

Our empirical results show that HuRL, when given the right scheduling of $\lambda$, improves even the performance of a well tuned SAC. Empirically, we found setting $\lambda$ around 0.95~0.98 usually leads to reasonably good performance, though it might not be the best environment-specific choice. In the paper, we do highlight that the $\lambda$ schedule is an extra hyperparameter HuRL adds, and is a limitation of our approach, -- e.g., in Section 7. Automatically adapting the $\lambda$ schedule is indeed important for future work. We will update the paper to be more explicit about this.

---

### Decision · Program_Chairs · 2021-09-27

**Decision:**

Accept (Poster)

**Comment:**

I thank the authors for their submission and active participation in the discussions. The majority of reviewers find this paper valuable, in particular they emphasized the novelty [bqji], a good theoretical framework [RNky] with new findings that admissibility consistency help RL [58wi], a comprehensive discussion of related literature [RNky,vrFt], and that the paper is well structured and well written [RNky,vrFt]. On the negative side, reviewer bqji voiced concerns about limited experimental validation. I believe the authors have addressed these concerns well in their rebuttal. I thus side with reviewers RNky, vrFt and 58wi, and recommend acceptance. I encourage the authors to further improve their paper based on the reviewer feedback.